


# Flood-Related Extreme Precipitation in Southwestern Germany: Development of a Two-Dimensional Stochastic Precipitation Model

Florian Ehmele[1] and Michael Kunz[1,2]

[1]Institute of Meteorology and Climate Research, Karlsruhe Institute of Technology (KIT), Hermann-von-Helmholtz-Platz 1, 76344 Eggenstein-Leopoldshafen, Germany.
[2]Center for Disaster Management and Risk Reduction Technology (CEDIM), KIT - Karlsruhe, Germany

*Correspondence to:* Florian Ehmele (florian.ehmele@kit.edu)

**Abstract.** Various application fields, such as insurance industry risk assessments for the design of flood protection systems, require reliable precipitation statistics in high spatial resolution, including estimates for events with high return periods. Observations from point stations, however, lack of spatial representativeness, especially over complex terrain, and do not reliably represent the heavy tail of the distribution function. This paper presents a new method for stochastically simulating precipitation fields based on a linear theory of orographic precipitation and additional functions that consider synoptically driven rainfall and embedded convection in a simplified way. The model is initialized by various statistical distribution functions describing prevailing atmospheric conditions, such as wind vector, moisture content, or stability, estimated from radiosonde observations for a limited sample of the 200 strongest rainfall events observed.

The model is applied for the stochastic simulation of heavy rainfall over the complex terrain of Southwest Germany. It is shown that the model, despite its simplicity, yields reliable precipitation fields. Differences between observed and simulated rainfall statistics are small, being in the order of only ±10% for return periods of up to 1,000 years.

## 1 Introduction

Persistent precipitation over large areas and the resulting widespread flooding frequently cause major damage in Central Europe in the order of several billion EUR. In Germany, two extreme floods in 2002 and 2013 with estimated return periods of more than 200 years (Schröter et al., 2015) collectively caused more than EUR 22 billion in economic losses (inflation adjusted to 2017; MunichRe, 2017). Besides these extreme events, smaller floods with higher frequencies, such as those in the years of 2005, 2006, 2010, and 2011 (Uhlemann et al., 2010; Kienzler et al., 2015), also contribute to the large damage potential associated with floods.

Flood risk estimation, for example, for insurance purposes or for the design of appropriate flood mitigation systems, requires the reliable statistical analysis of extreme rainfall. Traditionally, these extremes have been estimated at point stations from intensity-duration-frequency (IDF) with extreme value statistics being applied (Koutsoyiannis et al., 1998). This method, however, implies two caveats: (i) the low spatial representativeness of point observations and (ii) the limited observation period so that not all possible extreme configurations enter the samples. To account for the former point, either geostatistical interpolation routines, such as kriging (Goovaerts, 2000), or techniques that relate precipitation to both orographic characteristics and



atmospheric parameters (e.g., Basist et al., 1994; Drogue et al., 2002) are applied. Shortcomings resulting from these methods are the lack of representativeness of station data with respect to the surroundings, and the neglect of dynamical and thermo-dynamical processes decisive for real precipitation events. To account for the limited observation period, several studies have employed stochastic weather generators to simulate precipitation events at single grid points (e. g., Richardson, 1981; Furrer

and Katz, 2007; Neykov et al., 2014). A recent study by Cross et al. (2017) introduced a censored rainfall modeling approach designed to reduce the underestimation of extremes. Albeit considering the long-term variability of precipitation, which leads to more reliable estimates for extremes, these approaches still lack of spatial representativeness.

In the present study, we propose a two-dimensional stochastic precipitation model (SPM2D) that allows for simulating a large number of precipitation fields with high spatial resolution. Large sample sizes of several thousand events are required

to obtain reliable and robust estimates of the hazard for high recurrence periods, such as the one-in-200-year events that have to be considered by insurance companies. The core of our SPM2D is the diagnostic linear model approach for orographic precipitation according to Smith and Barstad (2004). The model considers wave dynamics in terms of the linearized equations of a stratified, non-hydrostatic flow over mountains (Smith, 1980). Input parameters are atmospheric flow quantities connected to precipitation, such as stability, moisture scaling height, precipitable water, or flow speed, all estimated from radiosoundings.

Additional internal free parameters, such as characteristic time scales for cloud water conversion and fallout, serve as calibration parameters. The Smith and Barstad (2004) model has been successfully applied in various regions, such as several locations in the United States (Barstad and Smith, 2005), Iceland (Crochet et al., 2007), Southwest Germany (Kunz, 2011), or Southern and Northern Norway (Caroletti and Barstad, 2010; Barstad and Caroletti, 2013). It is found that despite the fact that characteristic time scales and background precipitation may vary from one situation to another, simulations using fixed values for the free

parameters yield reliable and robust results. In our approach, we added two additional components to the orographic and background precipitation: synoptic-scale fronts and convection embedded into mainly stratiform clouds (Fuhrer and Schär, 2005). Whereas the former component may enhance (or reduce in case of absence) precipitation over larger areas, the latter may lead to slightly locally enhanced totals.

In the present study, we applied the SPM2D to both single events with heavy rainfall and over a long-term period of several

thousand years (events) over Southwest Germany (Fig. 1). In the latter case, the required model parameters are estimated from probability density functions (pdf) and are stochastically simulated. In this application, we fixed the internal free parameters to constant values estimated thorough calibration. Due to varying precipitation regimes in summer and winter, we seasonally differentiate our analyses.

The presented SPM2D is one component of a risk assessment methodology that estimates the risk for a local direct insurer

by quantifying the maximum probable loss for a 200-year return period (PML200). The other risk assessment components, however, are not further discussed in this paper.

The paper is structured as follows: Section 2 briefly describes the data sets used in this study. Section 3 introduces the basics of the SPM2D. Section 4 presents the results of the calibration based on a set of 200 heavy rainfall events, and Section 5 shows some characteristics of the selected events. Simulation results are discussed in Section 6, and Section 7 lists some conclusions.






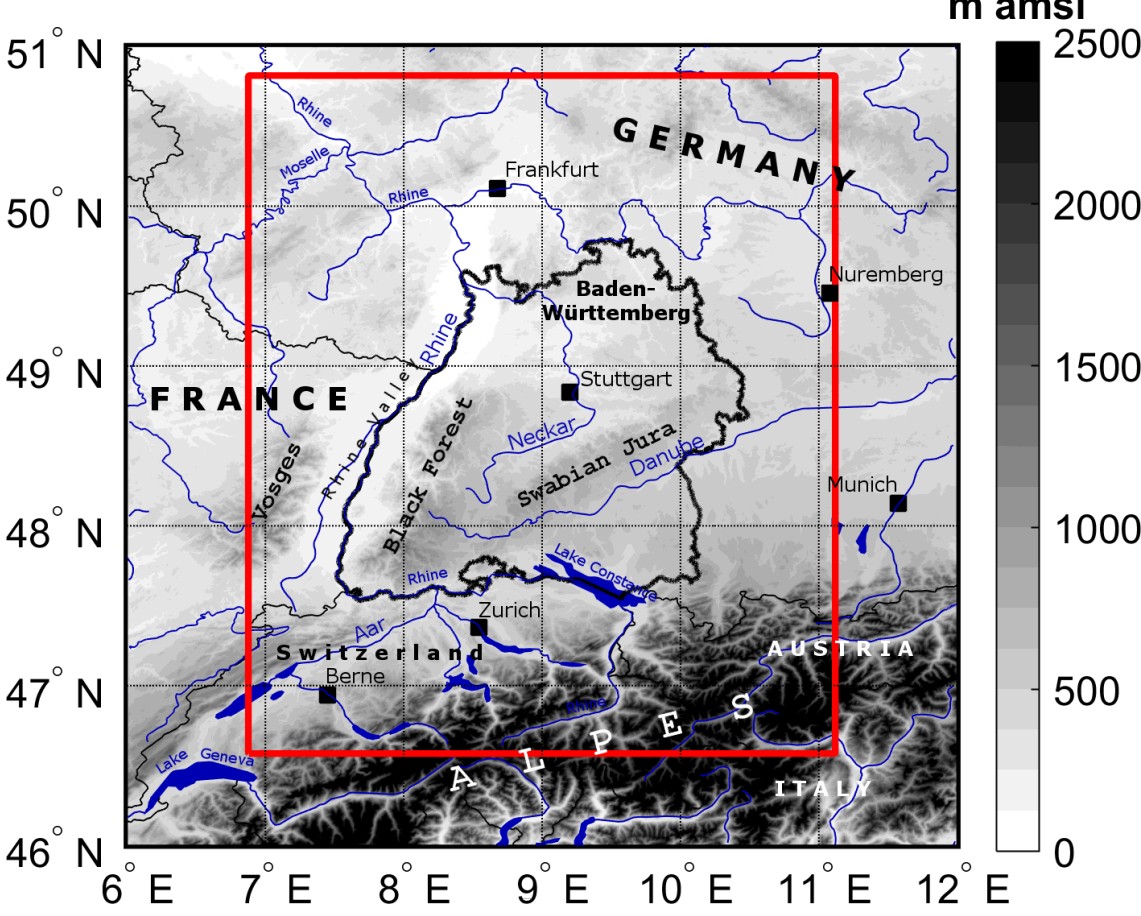

**Figure 1.** Topographic map of Southwestern Germany and surrounding areas with main river networks and lakes as well as substantial orographic structures; The national borders (slim solid black contours) and the border of the Federal State of Baden-Württemberg (bold solid black contour) are shown as well as the model domain (red box).

## 2   Data and methods

The SPM2D presented in this study is based on two different data sets: gridded precipitation data to estimate background precipitation and to calibrate and verify the model, and vertical profiles from radiosondes to initialize the model. Unless otherwise indicated, the investigation period covers the years of 1951–2016 (hereinafter referred to as IP). The investigation

5   area is the Federal State of Baden-Württemberg (BW) in Southwest Germany, which extends from 46.6 to 50.8°N and from 6.9 to 11.1°E (Fig. 1). The terrain exhibits a certain complexity with the broad Rhine Valley; with elevations of 100–200 m bounded by the Vosges Mountains (France) to the west and the Black Forest mountains to the east; with a maximum elevation



of 1493 m (Feldberg); and with some rolling terrain to the northeast. Annual precipitation is between 600 (southern Rhine Valley) and approximately 2000 mm (southern Black Forest).

## 2.1 Rainfall totals

Rainfall statistics in our study are based on the REGNIE (***REG****ionalisierte **NIE**derschläge*, regionalized precipitation) data set provided by the German Weather Service (Deutscher Wetterdienst; DWD). REGNIE is a gridded data set of 24-hour totals based on several thousand climate stations more or less evenly distributed across Germany (so-called RR collective). The REGNIE algorithm interpolates the observations to a regular grid considering elevation, exposition, and climatology (Rauthe et al., 2013). The REGNIE area contains 611 grid points in the west–east direction with $5.83°\text{E} \leq \phi \leq 16°\text{E}$ and 971 grid points in the north–south direction with $47°\text{N} \leq \theta \leq 55.08°\text{N}$ ($\phi$: longitude; $\theta$: latitude). Grid points outside of Germany are set to malfunction. The spatial resolution of REGNIE is approximately $1\,\text{km}^2$, and the observation period is from 06 to 06 UTC.

It should be noted that REGNIE data are temporally not homogeneous due to changes in the locations and number of rain gauges. Furthermore, because the number of stations considered by the regionalization is limited, especially over elevated terrain, such as the Black Forest mountains, areal precipitation exhibits a certain bias. Its magnitude, however, cannot be directly estimated from the observations solely (Kunz, 2011).

## 2.2 Radiosoundings

Input of the SPM2D are several atmospheric parameters derived from radiosoundings: thermal stability in terms of saturated Brunt-Väisälä frequency $N_\text{m}$ (e. g., Lalas and Einaudi, 1973) and actual and saturated vertical temperature gradients ($\gamma$ and $\Gamma_\text{m}$), water vapor scaling height $H_\text{w}$, water vapor mixing ratio $q_\text{v}$, wind speed $U$, and direction $\beta$ (see Sect. 3). These parameters are computed from the vertical profiles of temperature, moisture, wind speed, and direction at the radiosounding station of Stuttgart (48.83°N 9.20°E) located somewhat downstream of the northern Black Forest mountains. Even though the location might not be ideal because the profiles do not represent undisturbed conditions, the profiles in the mean are similar to that of the upstream station of Nancy in France as shown by Kunz (2011). Data from Nancy, however, are available after 1990 only and, thus, cannot be used in this study, whereas soundings from Stuttgart are available since 1957. In this study, we used the soundings at the main standard times for synoptic observations of 00 and 12 UTC.

Sounding data were provided by the Integrated Global Radiosonde Archive (IGRA) for quality-controlled radiosonde and pilot balloon observations from the National Climatic Data Center (Durre et al., 2006). These data at both main pressure levels and significant levels, where one of the parameters show a significant change, were interpolated into equidistant increments of $\Delta z = 10\,\text{m}$ (Mohr and Kunz, 2013). All parameters derived from the soundings refer to the lowest 5 km of the atmosphere since this layer is most relevant for air flow and stability. Furthermore, to account for the decreasing impact of higher atmospheric layers on the flow characteristics, all flow parameters $\Lambda$ have been vertically integrated ($\widetilde{\Lambda}$), with water vapor weighting being applied (Kunz, 2011):

$$\widetilde{\Lambda} = \frac{\int_{z=0}^{z_\text{t}} \Lambda \rho_\text{d} q_\text{v}\, dz}{\int_{z=0}^{z_\text{t}} \rho_\text{d} q_\text{v}\, dz}, \tag{1}$$



where $\rho_\mathrm{d}$ is the density of dry air and $z_\mathrm{t} = 5000\,\mathrm{m}$.

As some layers may be moist-unstable, resulting in imaginary $N_\mathrm{m}$, the averaging routine is applied to $N_\mathrm{m}^2$. In the few cases where $\widetilde{N}_\mathrm{m}$ was imaginary, it was set to a near-neutral, constant value of $0.0003\,\mathrm{s}^{-1}$.

### 2.3 Parameters for Embedded Convection

Embedded convection in the SPM2D is considered by single streaks of enhanced precipitation (see Sect. 3). These streaks are stochastically generated according to the statistical distributions of the observed maximum length $L$ and width $W$ of severe convective storms estimated by Fluck (2017). In that study, convective storms were identified from the constant altitude plan position indicator (CAPPI) for a reflectivity in excess of 55 dBZ, also known as the Mason (1971) criterion for hail detection. The application of a tracking algorithm based on the concept of the algorithm of TRACE3D (Handwerker, 2002) yields entire
tracks of convective storms. In total, more than 20,000 tracks over Germany, France, Belgium, and Luxembourg were identified during the summer half years (April to September) in the period 2004–2014. Even though we do not consider rainfall related to severe convective storms or hail in the SPM2D, the statistical distributions of the storm's dimensions are reliable proxies for the extension of enhanced precipitation from embedded convection.

### 2.4 Event definition and statistical distribution functions

Stochastic modeling of precipitation events with SPM2D requires the adjustment of appropriate probability density functions (pdf) to all input parameters. These pdfs are estimated from an appropriate set of past heavy rainfall events. Based on the pdfs, several thousands events can be stochastically generated (more details are found in Sect. 4). Because the characteristics of the ambient conditions and thus the precipitation regimes change throughout the year, we seasonally differentiate the estimated pdfs among spring (MAM), summer (JJA), autumn (SON), and winter (DJF).

In the first step, a sufficient and appropriate subset of relevant historic events was identified. An event here is defined as a period of one or more days with persisting precipitation above a certain threshold. Because our study focuses on major large-scale flood events and not on local-scale floods or flash floods, an extension to multi-day events is necessary. In this way, time delays in discharge response or flood waves traveling along river networks are implicitly considered (e. g., Duckstein et al., 1993; Uhlemann et al., 2010; Schröter et al., 2015).

We define the historic event set based on maximum areal precipitation. For this, we simply accumulate the (equidistant) 24-hour REGNIE totals $\overline{R}_\mathrm{BW}$ of all grid points in BW. Following the sorting of all values of $\overline{R}_\mathrm{BW}$ in descending order, the strongest 200 values enter the sample (top200). As precipitation is not limited to these (single) days but may be embedded in longer time periods, we define the threshold of $R_\mathrm{thres}$ for event definition. Estimating $R_\mathrm{thres}$, we consider "wet" days by using $\overline{R}_\mathrm{BW} > 0$ solely, and we set $R_\mathrm{thres}$ to the 75% percentile of this sample. A lower threshold leads to an over-interpretation of
longer clusters, a higher one avoids multi-day events.

Event precipitation starts on the first day that exceeds $R_\mathrm{thres}$. When areal means of consecutive days are also above $R_\mathrm{thres}$, they are simply accumulated, yielding events of more than one day. To ensure statistical independence, at least three days of non-exceedance have to prevail between two events in accordance with the approach of Palutikov et al. (1999) for wind storms.





On the day that $R_{\text{thres}}$ is not exceeded for the next three days as well defines the end of an event. In accordance with Piper et al. (2016), we solely count "rain days" ($\overline{R}_{\text{BW}} \geq R_{\text{thres}}$) and neglect "skip days" ($\overline{R}_{\text{BW}} < R_{\text{thres}}$) for event duration estimation, which is a widely used approach (Wanner et al., 1997; Petrow et al., 2009). This approach avoids the over-interpretation of longer clusters.

5    Based on the procedure described above, a defined precipitation event contains one or more days of the top200 sample.

In the next step, we identified the pdfs most appropriate for statistically describing each of the seven model parameters. In total, 17 different pdfs were tested and compared with the distribution functions of each parameter for each of the four seasons (Table 1). In addition to 20 pdfs preset by the MATLAB statistic toolbox (MATLAB, 2016), we considered the circular von-Mises distribution (Mardia and Zemroch, 1975) for wind direction only. Note that Gumbel (GbD) and Weibull (WbD)

10   distributions are special cases of the generalized extreme value distribution (GEV) and that some pdfs cannot be used for every parameter due to their ranges of validity.

To estimate the pdf that best fits the data, we estimated the appropriate number of histogram classes according to Freedman and Diaconis (1981), and we calculated the bias, root mean square error (rmse) and Spearman correlation coefficient $r_{\text{Sp}}$ (Spearman, 1904) as quality indicators (QIs). We also applied a $\chi^2$-test according to Wilks (2006) as a QI. For each QI, we

15   ranked the pdfs in ascending order and added up the rank numbers for each pdf receiving the best fit in terms of the least QI–rank sum (QIRS). In the case of the alikeness of two or more pdfs (about 10% of all cases), we manually selected the best one.

**Table 1.** List of the tested and suitable pdfs preset in the MATLAB statistical toolbox (the short acronyms in brackets are for further orientation).

| | |
|---|---|
| Birnbaum-Saunders (BSD) | Nakagami (NkD) |
| Gamma (GmD) | Normal (ND) |
| Generalized Extreme Value (GEV) | Poisson (PD) |
| Gumbel (GbD) | Rayleigh (RyD) |
| Half-Normal (HND) | Rician (RcD) |
| Inverse Gaussian (IGD) | Stable (SD) |
| Logistic (LD) | Student's t (StD) |
| Log-Logistic (LLD) | Weibull (WbD) |
| Log-Normal (LND) | |



# 3 Stochastic Precipitation Model

## 3.1 General description

The SPM2D, designed for widespread precipitation from essentially stratiform clouds, quantifies total precipitation $R_{\text{tot}}$ from the linear superposition of four processes and terms:

$$R_{\text{tot}} = R_{\text{oro}} + R_{\infty} + R_{\text{front}} + R_{\text{conv}} . \tag{2}$$

$R_{\text{oro}}$ estimates orographic rain enhancement, representing the central core of the SPM2D for complex terrain, such as those in BW. $R_{\infty}$ is the background precipitation related to large-scale lifting. These two parts originate from the linear orographic precipitation model of Smith and Barstad (2004) and Barstad and Smith (2005) with a few modifications, hereinafter referred to as reduced SPM2D (rSPM). In an extension of the rSPM, we included two additional precipitation components: $R_{\text{front}}$ to account for precipitation related to synoptic fronts, and $R_{\text{conv}}$ to consider embedded convection atop mainly stratiform clouds (e.g., Fuhrer and Schär, 2005; Kirshbaum and Smith, 2008). These two components were included because linear theory assumes waves that penetrate through the whole atmosphere, leading to an overestimation of precipitation totals, whereas at the same time, low intensities are underestimated (e.g., Kunz, 2011).

The SPM2D presented in this paper is an event-based model. Instead of simulating continuous long-term periods of several years, a specific number $n$ of independent events with various durations $t_{\text{ev}}$ occurring during different seasons is simulated. Since the purpose of the model is to stochastically simulate a large number of several thousands events, the results can be used to robustly estimate rare events, such as the one-in-200-year events that the insurance industry must consider (probable maximum loss, PML200). The prerequisite, however, is a reliable simulation of single events.

## 3.2 Orographic precipitation

The linear precipitation model of Smith and Barstad (2004) and Barstad and Smith (2005), which is briefly described in this subsection, is a simple yet efficient way to compute precipitation over complex terrain. A total number of only seven atmospheric parameters estimated from sounding data (see 2.2) are required to run the model. It is based on three-dimensional (3D) linear flow according to Smith (1980) and Smith (1989). Thus, it explicitly considers linear flow effects evolving over mountains, such as upstream-tilted gravity waves or flow that goes around rather than over an obstacle in the case of low wind speed, high static stability, and/or large mountains. It is assumed that saturated lifting produces condensed water that falls to the ground after a certain time shift (Jiang and Smith, 2003). Thus, precipitation on the ground is directly related to the condensation rate.

One of the key components of the linear model is a pair of linear steady-state equations for the advection of vertically integrated cloud water and hydrometeor density, $q_{\text{c}}$ and $q_{\text{h}}$, during characteristic time scales:

$$\mathbf{v} \cdot \nabla q_{\text{c}} = S(x,y) - \frac{q_{\text{c}}}{\tau_{\text{c}}} , \tag{3}$$

$$\mathbf{v} \cdot \nabla q_{\text{h}} = \frac{q_{\text{c}}}{\tau_{\text{c}}} - \frac{q_{\text{h}}}{\tau_{\text{f}}} , \tag{4}$$





where $\tau_c$ and $\tau_f$ are time scales for cloud water conversion and the fallout of hydrometeors respectively. Both time scales are mathematically analogous and are assumed to be constant in time and space. When the time scales are set to zero, the maximum precipitation is almost one order of magnitude larger compared with a configuration with, for example, $\tau_f = \tau_c = 1000\,\text{s}$ (Kunz, 2011). Source term $S$ describes the mass flux of precipitation caused by orographic lifting. For positive $S$, term $q_c\tau_c^{-1}$ acts as

a source in (4) and as a sink in (3). This term is proportional to the cloud water density integrated vertically from the bottom to the top of the lifting area. In light of this, it is assumed that the whole column is saturated in the case of lifting. The loss of hydrometeors, $q_h\tau_f^{-1}$ in (4), determines precipitation rate $R$ and is proportional to the hydrometeor column density. However, in the case of descending air with negative $S$ downstream of mountains, evaporation occurs, and $R$ may become negative.

A powerful method for solving the advection equations for cloud physics, (3) and (4), together with the linear theory for

3D flow is to apply a two-dimensional (2D) Fourier transform. In the Fourier space, precipitation rate $\hat{R}(k,l)$ is given by the following transfer function:

$$\hat{R}(k,l) = \frac{iC_w\sigma\hat{h}(k,l)}{(1 - imH_w)(1 + i\sigma\tau_c)(1 + i\sigma\tau_f)},\tag{5}$$

which relates the precipitation field in the Fourier space, $\hat{R}(k,l)$, and the orography, $\hat{h}(k,l)$, with the horizontal wavenumbers $(k,l)$. In Equation (5), $i$ is the imaginary unit, and $C_w = \rho_{S_{ref}}\Gamma_m\gamma^{-1}$ is the uplift sensitivity related to condensation rate

$\rho_{S_{ref}} = \rho_d q_v$, where $\rho_d$ is the density of dry air and $q_v$ the water vapor density, and where $\Gamma_m$ and $\gamma$ are the moist adiabatic and actual lapse rates respectively. Water vapor scale height $H_w$ is the height above ground where the vertical integrated horizontal water vapor flux has reached $e^{-1}$ of its ground value. $\sigma = Uk + Vl$ is defined as the intrinsic frequency with components $U$ and $V$ of the undisturbed horizontal wind vector that is assumed to be constant through time and space.

Whereas the nominator of (5) gives the dependency of vertical motion and orography, the first bracket of the denominator

describes the relation of the source term to airflow dynamics. The second and third terms of the denominator consider the advection of hydrometeors during characteristic time scales $\tau_x$ ($x = c; f$) and, in the case of descent, evaporation.

Vertical wavenumber $m$ in (5) is given by the dispersion relation (Smith, 1980):

$$m(k,l) = \left[\frac{N_m^2 - \sigma^2}{\sigma^2}(k^2 + l^2)\right]^{0.5}.\tag{6}$$

In this formulation, $m$ controls both the depth and tilt of forced ascent or descent. Because vertical lifting is assumed to be

saturated throughout the whole column, saturated Brunt-Väisälä frequency $N_m$ has to be considered instead of the dry one, $N_d$. Compared with unsaturated flow, saturated flow leads to a weakening of the amplitude of the gravity waves via the reduction of static stability and thus to a flow that goes more directly over the mountains rather than around as shown, for example, by Durran and Klemp (1982) or Kunz and Wassermann (2011). Even though the concept of saturated flow by simply considering $N_m$ must be regarded as an approximation of the reality, it has been successfully applied by several authors studying flow

dynamics and precipitation (Jiang and Smith, 2003; Smith and Barstad, 2004; Kunz and Wassermann, 2011).

The precipitation field on the ground is obtained via an inverse Fourier transform of the transfer function (5):

$$R_{oro}(x,y) = \iint \hat{R}(k,l)e^{i(kx+ly)}\mathrm{d}k\mathrm{d}l.\tag{7}$$





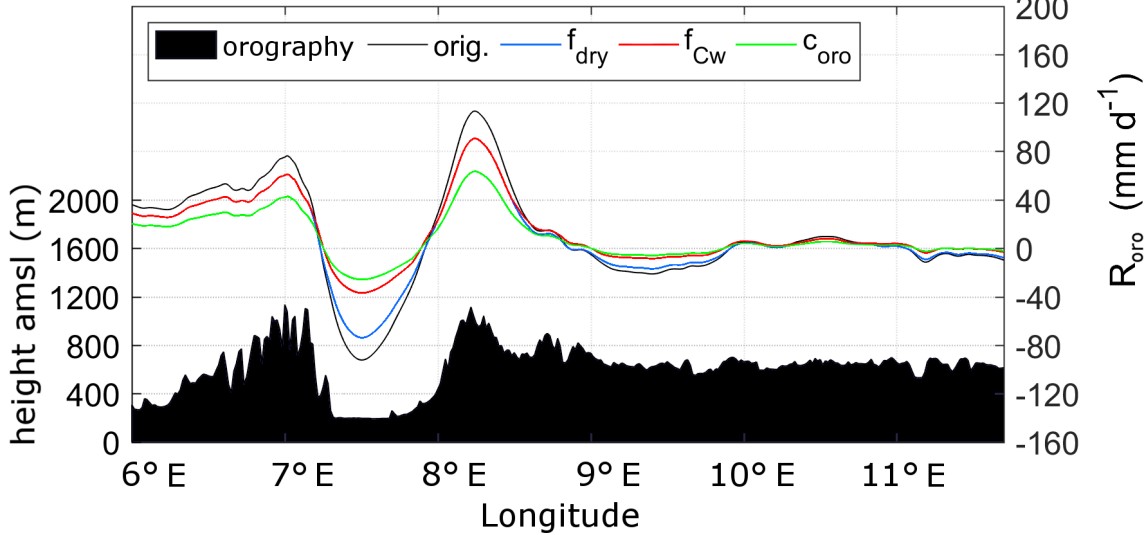

**Figure 2.** Different effects of the implemented internal free parameters $f_{\mathrm{dry}}$ (blue), $f_{\mathrm{C_w}}$ (red) and $c_{\mathrm{oro}}$ (green) on the original orographic precipitation part (black curve) for a west to east cross section through the model domain. The underlying orography is shown in black.

Note that even though $R_{\mathrm{oro}} < 0$ might be mathematically possible, negative total precipitation does not make sense physically and thus is truncated away. Therefore, we set $R_{\mathrm{tot}}(x,y) = \max(R_{\mathrm{tot}}(x,y), 0)$.

The model has five internal free parameters that can be used to adjust/calibrate the model to the observations. Three of these parameters are implicitly considered in the transfer function (Eq. 5): the two time scales of $\tau_{\mathrm{c}}$ and $\tau_{\mathrm{f}}$, which, however, are virtually identical, and the uplift sensitivity factor $C_{\mathrm{w}}$. The latter is modified with a multiplier to $C_{\mathrm{w}}^* = f_{\mathrm{C_w}} \cdot C_{\mathrm{w}}$ with the new factor $C_{\mathrm{w}}^*$ replacing the original $C_{\mathrm{w}}$ in (5). $f_{\mathrm{C_w}}$ reduces the sensitivity of the model for lifting, and therefore, the precipitation rate is reduced, especially over mountainous terrain (Fig. 2, red curve). The used model formulation allows for multiple ascents/descents of a virtual air parcel without any change in its water vapor content. Actually, water vapor is partly removed due to condensation processes during ascent, which is realized by $f_{\mathrm{C_w}}$.

An additional parameter, $f_{\mathrm{dry}}$, is implemented to reduce evaporation in descent regions, where $R_{\mathrm{oro}} < 0$ (Fig. 2, blue curve). The resulting underestimation of precipitation is found especially downstream of steeper mountains with greater descent (Kunz, 2011). Therefore, we implemented a new (multiplicative) parameter $f_{\mathrm{dry}}$ in (7), which is $f_{\mathrm{dry}} < 1$ only at grid points $(x,y)$ where $R_{\mathrm{oro}} < 0$; and $f_{\mathrm{dry}} = 1$ in all other cases.

Finally, the last and additional calibration parameter, $c_{\mathrm{oro}}$, reduces orographic precipitation in the whole domain (Fig. 2, green curve). It is a consequence of the assumption that the vertical lifting of an entire air column with saturation produces condensate and fallout at any time. In reality, not all layers are completely saturated, and water may also partly be stored by clouds. Parameter $c_{\mathrm{oro}}$ is implemented similarly to $f_{\mathrm{dry}}$ in Equation (7) but is constant for the whole area. With these two



parameters, orographic precipitation is modified to:

$$R_{\mathrm{oro}}^{*}(x,y) = f_{\mathrm{dry}} \cdot c_{\mathrm{oro}} \cdot R_{\mathrm{oro}}. \tag{8}$$

Note again that $f_{\mathrm{dry}}$ affects only grid points with net descent, whereas $c_{\mathrm{oro}}$ is constant over the whole domain.

### 3.3 Background precipitation

Background precipitation term $R_{\infty}$ (Eq. 2) describes the effect of large-scale lifting by synoptic-scale weather patterns. According to the $\omega$-equation, lifting is the result of three different mechanisms: positive vorticity advection increasing with height (or vice versa); the maximum of diabatic phase transitions; and the maximum of warm air advection. Even though lifting is the superposition of these three mechanisms, it does not make sense to split $R_{\infty}$ accordingly. Furthermore, we assume that the large-scale conditions are almost horizontally homogeneous across the investigation area, and so is $R_{\infty}$ at each time step.

To simplify the inclusion of large-scale lifting in the SPM2D, we estimate $R_{\infty}$ from REGNIE totals over a larger area with almost flat terrain, where $R_{\mathrm{oro}}$ as well as evaporation by ascent to a large degree are minimized. However, an analysis of various past events show the strong variability of the spatial distribution of precipitation even over flat terrain. For example, some events affect only the northern parts of the investigation area, whereas others only the southern parts. To ensure the proper estimation of $R_{\infty}$, we choose an area that covers most of the total investigation area but excludes the Black Forest and

prealpine lands. In the region, where we estimate $R_{\infty}$ (Fig. 3, black box), heavy rainfall is very unlikely. Totals of more than 50 mm per day, for example, exhibit an annual exceedance probability $p$ of less than 0.5. Furthermore, as confirmed by Figure 3, the probability of rain totals in excess of 50 mm per day is more or less homogeneously distributed. On average over 66 years, it can be assumed that precipitation in the area used for $R_{\infty}$ estimation mainly result from large-scale lifting and to a lesser extent from orographic influences.

### 3.4 Frontal precipitation

Apart of large-scale lifting connected to low-pressure systems or waves in the flow patterns, precipitation is also substantially enhanced by weather fronts. Active fronts may increase precipitation considerably due to cross-frontal circulations and lifting in the warm sector of a cyclone (e. g., Bergeron, 1937; Eliassen, 1962). Conversely, if a font affects only parts of the investigation area (e.g., a trailing front, where the flow is almost parallel to the frontal alignment), regions outside the sphere of influence

may experience much less or even no rain at all. Both effects are considered by implementing an additional quantity $R_{\mathrm{front}}$ in (2):

$$R_{\mathrm{front}} = (R_{\mathrm{oro}} + R_{\infty}) \cdot (c_{\mathrm{front}} - 1), \tag{9}$$

where $c_{\mathrm{front}}$ serves as the enhancement or reduction factor of the overall precipitation. In this simple parameterization, $R_{\mathrm{oro}}$ is considered again because frontal precipitation is additionally enhanced by orography as shown, for example, by Browning

et al. (1975) or Houze and Hobbs (1982). Due to the additive superposition of all precipitation components in (2), we have to subtract the original precipitation totals leading to a total multiplier $(c_{\mathrm{front}} - 1)$. The frontal enhancement factor is a function



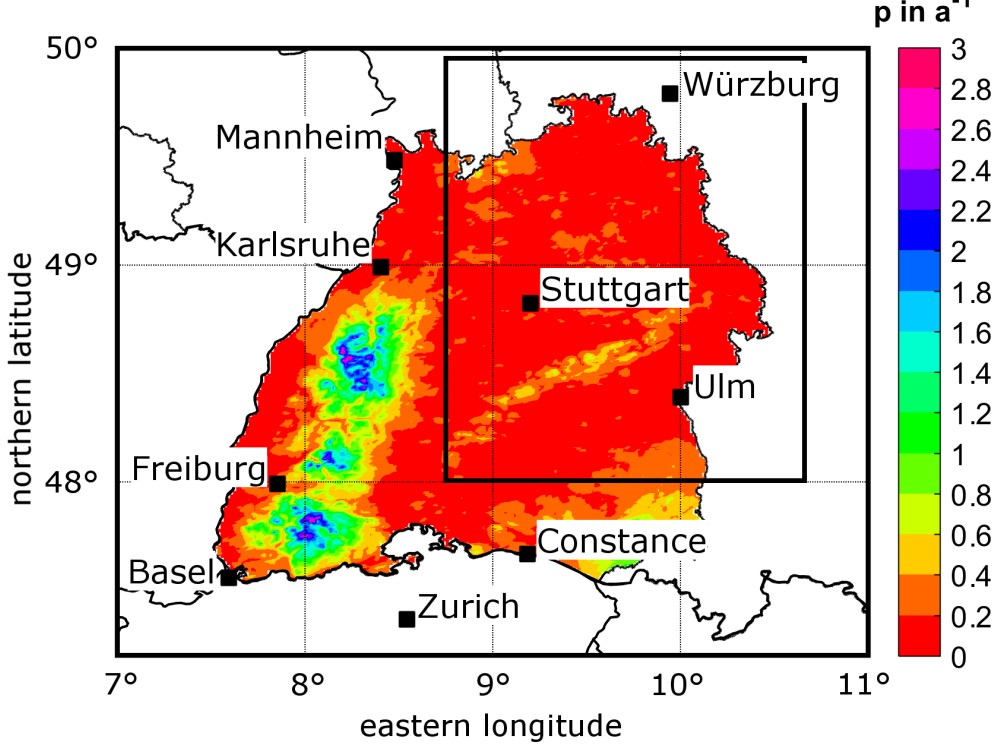

**Figure 3.** Probability of 24-hour REGNIE totals greater than 50 mm expressed as the average days per year for Baden-Württemberg; the black box indicates the area, where background precipitation $R_\infty$ is estimated.

of space realized by a rectangular area $c_{\text{front}}(x,y)$ (Fig. 4), where the orientation of the $y$-axis is prescribed by mean wind direction $\beta$.

To avoid strong gradients at the border areas of the rectangular, we applied Gaussian-shaped smoothing. Along the $x$-dimension, the spread is set to $8\sigma_{\text{n}}$, where $\sigma_{\text{n}}$ is the standard derivation of the normal distribution. In the $y$-direction, an infinitesimal length is considered (Fig. 4). As the minimum of $c_{\text{front}}$ is zero, $R_{\text{front}}$ can also attain negative values, thus leading to a weakening of total precipitation in an area affected or not affected by a front. To calculate $c_{\text{front}}$ from the observational data, we define this quantity as the relative difference between observations $O$ (REGNIE) and output $M$ of the rSPM (neglecting embedded convection as described in the next paragraph). This is expressed by

$$c_{\text{front}} = \overline{O} \cdot \overline{M}^{-1} \qquad (10)$$

assuming that the differences originate primarily from frontal effects. For the quantification of $c_{\text{front}}$, we use spatial mean values over the investigation area $\overline{O}$ and $\overline{M}$. The corresponding pdf for stochastic modeling is estimated using the least QIRS method with seasonal differentiation.



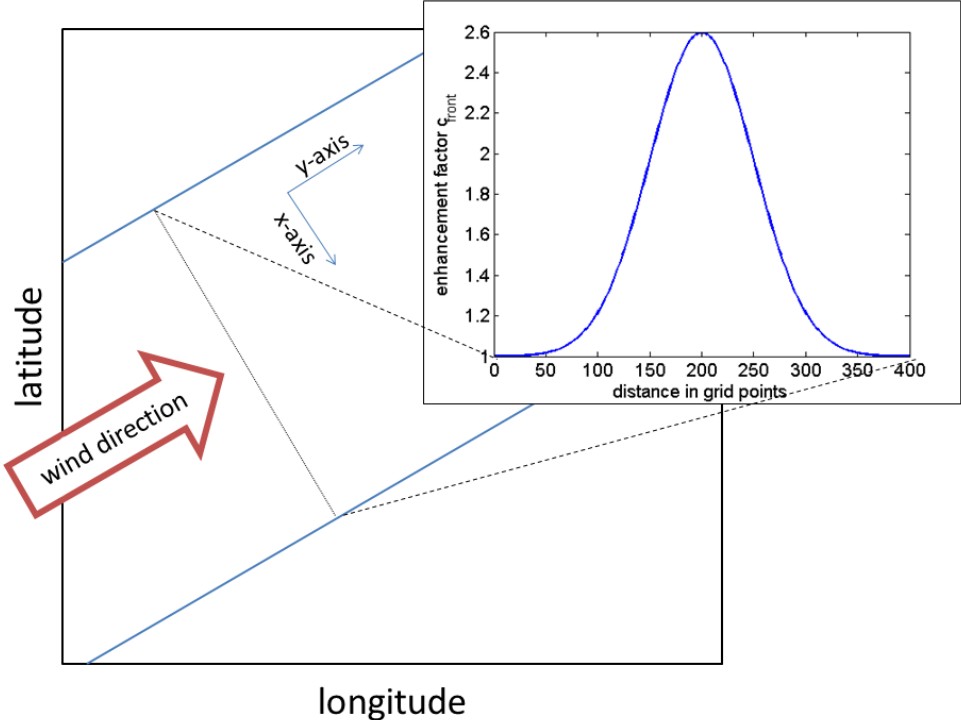

**Figure 4.** Schematic of a Gaussian-shaped distribution of the frontal enhancement factor with $c_{\mathrm{front}} = 2.6$ and $\sigma_{\mathrm{n}} = 50$ (upper right corner) and its location in the model domain for a southwesterly wind direction (arrow). The blue lines indicate the boundaries of the frontal zone.

### 3.5 Embedded Convection

The last part of the total precipitation model SPM2D considers convection embedded in mainly stratiform clouds (e. g., Fuhrer and Schär, 2005). Note, however, that the model is not foreseen to simulate purely convection. Such embedded convection mainly occurs when lifting is locally enhanced at mid- and upper tropospheric levels leading to a decrease of thermal stability

5 by the release of the latent heat of condensation (e. g., Kirshbaum and Durran, 2004; Kirshbaum and Smith, 2008; Cannon et al., 2012). Convection in general involves several complex processes that make simulation a difficult task. Since our model is restricted to large-scale precipitation with the objective of quantifying extremes in terms of areal precipitation solely, we treat embedded convection in a very simplified way by implementing several rectangular cells similar to the approach of frontal system consideration.

10 Because embedded convection is also partly induced by orographic precipitation mechanisms, we implemented a multiplicative factor to the precipitation fields related to both orographic and large-scale lifting, similar to the frontal part:

$$R_{\mathrm{conv}} = c_{\mathrm{conv}} \cdot (R_{\mathrm{oro}} + R_{\infty}), \tag{11}$$

with enhancement factor $c_{\mathrm{conv}}$.





For each time step of the simulation, we choose a number of convective cells, each with specific width $W$ and length $L$, and distribute these randomly over the whole model domain (Fig. 5). Both width $W$ and length $L$ of each rectangle of the convective cells are estimated from the characteristics of the severe convective storms identified from radar data by Fluck, 2017 (see Sect. 2.3). Furthermore, we restricted the two parameters to $L > W$ and $L_{\max} = 300\,\mathrm{km}$, or 300 grid points, respectively.

As for the frontal systems, the wind direction defines the orientation of the longer sides of the rectangles. For each convective cell, we choose $L \cdot W$ specific factors $c_{\mathrm{conv}}$ with $c_{\mathrm{conv}} \in \{0; 1\}$. As found, for example, by Fuhrer and Schär (2005) or Cannon et al. (2012), embedded convection can enhance precipitation up to 200%; thus, the given range of $c_{\mathrm{conv}}$ is adequate. Within the single cells, the spatial distribution of $c_{\mathrm{conv}}$ randomly varies between the given borders. Summing up all cells enables more than one cell per day at a specific grid point. The complete convective precipitation field for each time step is spatially

smoothed to avoid sharp gradients. Opposite of the Gaussian shape smoothing due to a more or less continuous ascent/descent of precipitation enhancement in the case of fronts, we use a moving average with a span of 10 grid points to preserve the high spatial variability of convection.

## 4   Calibration

This section describes the calibration of the SPM2D by comparing modeled and observed precipitation fields (REGNIE 24-hour

totals). The combination of the free parameters with the highest skill of the simulated rainfall totals is used for the stochastic simulations of 10,000 rainfall events, which is equivalent to a period of several thousand years as described in Section 6.

### 4.1   Method

The free model (calibration) parameters, $\tau^*$, $f_{\mathrm{C_w}}$, $f_{\mathrm{dry}}$ and $c_{\mathrm{oro}}$, are assessed based on the event set of top200. All other parameters required by the SPM2D (cf. Sect. 3) are quantified from radiosounding profiles at Stuttgart. In this evaluation, the

stochastic components of the SPM2D and the randomly modeled components for fronts ($R_{\mathrm{front}}$) and embedded convection ($R_{\mathrm{conv}}$) are neglected. Without these components, the model is referred to as the reduced SPM2D (rSPM).

   To determine appropriate values of the free parameters, a large number of model simulations was carried out with the rSPM. Whereas one parameter was successively varied, the others were kept constant. The selected ranges and increments of the parameters listed in Table 2 resulted in 2,016 possible parameter combinations, giving a total number of approximately

390,000 simulation days for the top200 event set. For each day and parameter combination, we assess the model skill by quantifying both bias and rmse. Both data sets (model output and REGNIE) are slightly smoothed using a running 5×5 grid box. The reason for the smoothing is that REGNIE data, despite having a high resolution of 1 km, exhibit spatial uncertainty due to the limited number of observational data considered. Especially around the crests of Black Forest, where the number of stations is very low, REGNIE data cannot reproduce local peak rainfall totals. Furthermore, as shown, for example, by Barstad

and Smith (2005), smoothed data yield more robust results when comparing model and observation data. Note, however, that larger values for $\tau^*$ and smaller values of $f_{\mathrm{C_w}}$, respectively, likewise smooth the simulated precipitation fields. In these cases, the QIRS method used for the evaluation (Sect. 2.4) has to be applied carefully.





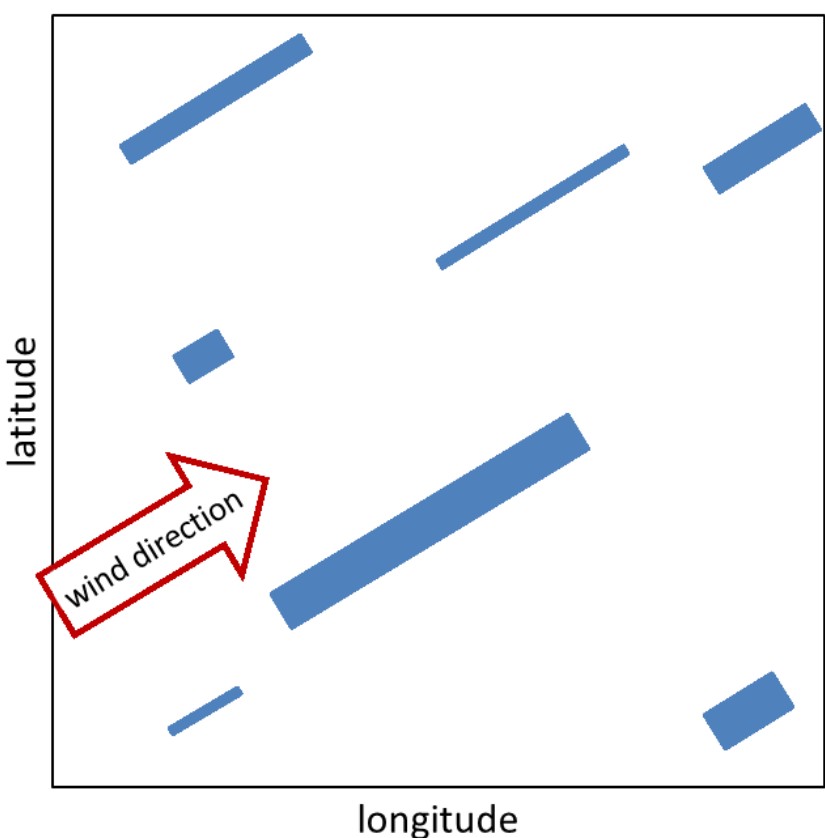

**Figure 5.** Schematic of the convection implementation with rectangular cells (blue). The orientation is defined by the wind direction (arrow); each cell is assigned to an individual factor $c_{\mathrm{conv}}$.

**Table 2.** The minimum and maximum values, and the increments of the time scales $\tau^*$, and multiplicative factors for the uplift sensitivity $f_{\mathrm{C_w}}$, the lee-side drying $f_{\mathrm{dry}}$, and the adjustment of orographic precipitation $c_{\mathrm{oro}}$.

| parameter | minimum | maximum | increment |
|:---:|---:|---:|---:|
| $\tau^*$ | 800 s | 1500 s | 100 s |
| $f_{\mathrm{C_w}}$ | 0.5 | 1.0 | 0.1 |
| $f_{\mathrm{dry}}$ | 0.4 | 1.0 | 0.1 |
| $c_{\mathrm{oro}}$ | 0.5 | 1.0 | 0.1 |





To avoid apparently better representations of smoothed data fields, we use skill score $S$ (Eq. 12) described by Taylor (2001) for evaluating climate models to determine the best parameter combination of rSPM:

$$S = \frac{4\,(1+r)}{\left(\hat{\sigma}_f + \frac{1}{\hat{\sigma}_f}\right)^2 \cdot (1+r_0)}, \tag{12}$$

where $r$ is the correlation coefficient after Spearman (1904), $r_0$ the maximum attainable correlation, and $\hat{\sigma}_f = \sigma_{\mathrm{mod}} \cdot \sigma_{\mathrm{obs}}^{-1}$ the normalized standard deviation with the standard deviation of model output $\sigma_{\mathrm{mod}}$ and that of observations $\sigma_{\mathrm{obs}}$. For $\hat{\sigma}_f \to 1$ and for $r \to r_0$, $S$ approaches unity, which is the best result. According to Taylor (2001), improved values of rmse or bias does not mean an actual improvement of the model performance, and the use of correlation and standard deviation is more stable. Furthermore, Taylor (2001) provided no regulation for the estimation of $r_0$. Therefore, we set $r_0$ to the maximum calculated correlation coefficient of all simulations. As it is not guaranteed that this maximum is the actual maximum attainable correlation, we increase $r_0$ by 10%, which yields $r_0 = 0.93$.

Skill score $S$ is computed for each simulation day and each parameter combination. From all realizations, we select the parameter combination that yields the highest median value of $S$ averaged over all top200 events.

## 4.2 Calibration Results

Applying the method to the top200 events as described above, the highest median skill score of $S = 0.60$ is obtained for the combination of $\tau^* = 1400\,\mathrm{s}$, $f_{\mathrm{C_w}} = 1.0$, $f_{\mathrm{dry}} = 0.4$ and $c_{\mathrm{oro}} = 0.8$. For this combination, the other median values of the quality indices are $r_{\mathrm{Sp}} = 0.39$, $\hat{\sigma}_f = 0.98$, bias $= 6.30\,\mathrm{mm}$, and rmse $= 14.85\,\mathrm{mm}$. The assessed values for the former two model parameters are physically plausible and comparable to other studies with the rSPM (e.g., Barstad and Smith, 2005; Caroletti and Barstad, 2010; Kunz, 2011). The latter two parameters are incorporated exclusively in this study. However, considering the slight overestimation of orographic precipitation enhancement and the strong overestimation of lee-side drying, the two values are also physically plausible.

The sensitivity of skill score $S$ to $\tau$ and of the two other parameters, $f_{\mathrm{C_w}}$ and $c_{\mathrm{oro}}$ (Figure 6), shows a dipole structure in both cases with the highest values of $S$ along a counter diagonal. Minor skill scores obtain with the shortest (longest) time scales in combination with the highest (lowest) uplift sensitivity or highest (lowest) weighting of $R_{\mathrm{oro}}$ in Equation 2. This implies, on the one hand, that for smaller displacements of precipitation from the formation region, orographic precipitation is overestimated by the rSPM and thus has to be reduced. On the other hand, $R_{\mathrm{oro}}$ has to increase for wider displacements.

Note, however, that the above-identified parameter combination yields the lowest errors only when averaging over all events. Single events may become more realistic with another parameter combination, reflecting particularly the unknown, and thus not considered microphysical processes that are decisive for precipitation formation and that are strongly controlled by vertical wind speed, temperature, and moisture profiles. The dependency of microphysical processes on ambient conditions, however, is not relevant when running the model in the stochastic mode as in this study.



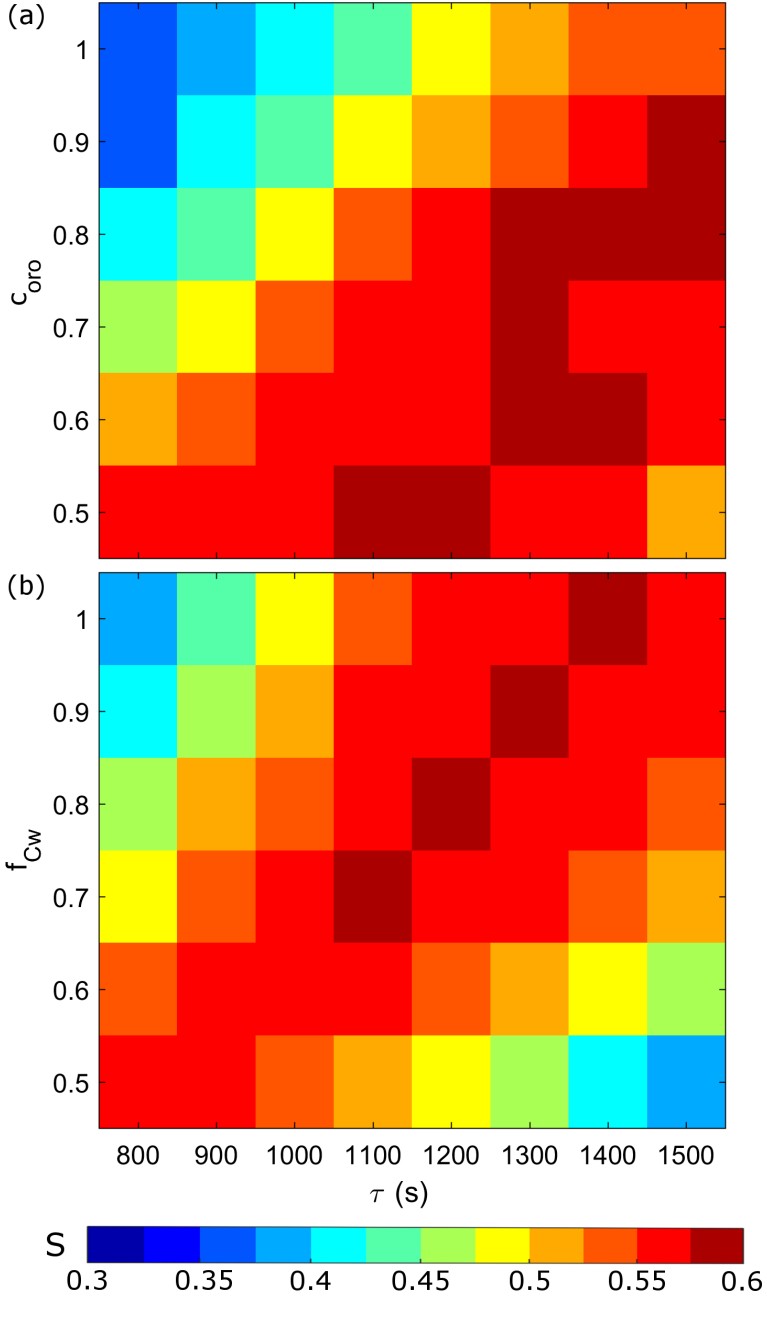

**Figure 6.** Sill score $S$, averaged over the top200 event set, depending on $\tau$ and (a) $c_{\mathrm{oro}}$, and (b) $f_{\mathrm{C_w}}$, while the other free parameters, respectively, were set to their optimum values.





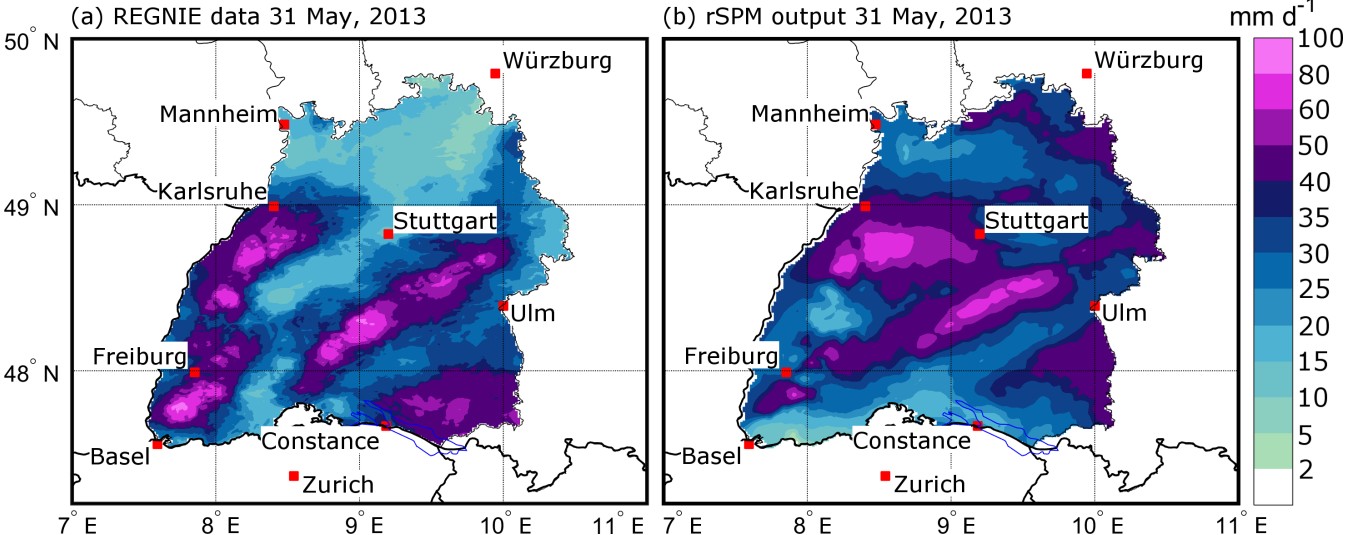

**Figure 7.** Comparison of (a) REGNIE 24-hour rainfall totals, and (b) rSPM output for Southwest Germany, exemplary on 31 May, 2013. Note that REGNIE data are available for Germany only. The parametrization in (b) is $\tau^* = 1400\,\mathrm{s}$, $f_{C_w} = 1.0$, $f_{dry} = 0.4$, and $c_{oro} = 0.8$. The areas outside of Baden-Württemberg are covered white for better visualization and comparison.

### 4.3  Case Study

After the parameter adjustment, the rSPM tends to slightly underestimate orographic precipitation, whereas totals over flat or rolling terrain are overestimated. This behavior can also be seen in the example of 31 May 2013 (Fig. 7), a heavy precipitation event that trigged the severe flooding in 2013 (Schröter et al., 2015).

5    On that day, a pronounced low pressure with its center over Croatia led to the sustained advection of moist airmasses from northerly directions around 20° in combination with synoptic-scale ascent. The Stuttgart sounding with low stability ($N_m = 0.0055\,\mathrm{s}^{-1}$), high precipitable water ($pw = 24\,\mathrm{kg\,m}^{-2}$), and high wind speed ($U = 20\,\mathrm{m\,s}^{-1}$), the latter two determining the horizontal water vapor flux, is already an indication of high precipitation totals, especially over the Northern Black Forest. Consequently, precipitation totals across the investigation area were between 10 and 100 mm.

10    Overall, the rSPM is able to reproduce most of the structures of the observed rain field (Fig. 7). The quality indices for that day are $S = 0.62$, $r_{Sp} = 0.30$, $\hat{\sigma}_f = 0.75$, bias $= 4.44\,\mathrm{mm}$, and rmse $= 14.82\,\mathrm{mm}$. The best agreement between observed and simulated precipitation fields is found for the Northern Black Forest as well as Swabian Jura. Over the northern part of the model domain (north of 49° N) and southwest of Stuttgart, simulated rainfall is substantially higher compared with REGNIE. By contrast, the rSPM simulates lower totals in the Southern Rhine Valley near and over the mountainous regions 15 of the Southern Black Forest (around Freiburg), especially east of the Basel region, where lee-side evaporation in the model dominates.





One reason for the discrepancy between observed and simulated precipitation might be the improper location of the Stuttgart sounding used for the model initialization. Since orographic precipitation in the rSPM strongly depends on the initial conditions of the used sounding data, which may be affected by the upstream terrain of northeastern Baden-Württemberg, we conduct a sensitivity study of this event by varying the ambient conditions. Following Kunz (2011), we perturbed the estimated values

for $N_{\mathrm{m}}^2, q_{\mathrm{v}}, U$, and $\tau$ by multiplicative factor $var\_mult$ increasing linearly from 0.5 to 2.0 in increments of 0.1. Wind direction $\beta$ was varied in the range of $\pm 30°$ in increments of $5°$.

The best results in terms of the lowest rmse (Figure 8) are obtained for higher stability (increase of $N_{\mathrm{m}}^2$) or longer time scales, whereas in the case of water vapor density $q_{\mathrm{v}}$ or horizontal wind speed $U$, the lowest rmse is obtained when decreasing the original values. The results also reveal a higher sensitivity of the rSPM to changes in water vapor and wind speed compared

with stability or microphysical time scales. Regarding wind direction $\beta$, only small changes can be detected for $\Delta\beta$ between $-30°$ and $+10°$ with the lowest rmse for the original value. The main reason for this is the orientation of the major orographic structures (e. g., the Black Forest) from southwest to northeast, for which variations of $\beta$ become relevant only for more easterly shifts ($\Delta\beta > 10$), resulting in a steeper inflow angle.

The highest skill score $S$, conversely, is reached for increasing $U$ and $q_{\mathrm{v}}$, and decreasing $\tau$ and $N_{\mathrm{m}}^2$. In the case of wind

direction, $S$ continuously decreases from 0.8 in the northwesterly inflow to 0.4 in the northeasterly winds.

For the case study of 31 May 2013, the observed mean for Baden-Württemberg is $\bar{R}_{\mathrm{obs}} = 33.1$ mm, whereas the simulated mean is $\bar{R}_{\mathrm{mod}} = 37.3$ mm, only 12.6% higher compared with the observations. The rmse and skill score $S$ are near the optimum when perturbing different variables. The deviations of spatial means and quality indices are at a reasonable level. However, as already explained, the SPM2D is not designed to represent historic events in detail. Other parameter combinations may yield

even better results for this single event.

## 5    Parameter estimation for the stochastic simulations

### 5.1    Adjustment of the distribution functions

Stochastic model simulations are based on pdfs that are adjusted to the required parameter. Event duration, background and frontal precipitation as well as preconditions are estimated from REGNIE data for the top200 event set. Ambient parameters

required by the SPM2D are derived from vertical profiles of the radiosondes at Stuttgart, whereas the width of embedded convection is estimated from the radar tracks of severe convection. Furthermore, as mean ambient conditions and thus precipitation characteristics change throughout the year, we differentiate among the four seasons.

After separating the historic event set into the four main seasons, we estimate for each of the 10 parameters the pdf that best fits the distribution of the observations ($= 10$ parameters $\times$ 4 seasons $= 40$ cases; Table 3) by using the least QIRS method

(cf. 2.4). From the overall 21 pdfs that were considered, only 12 are suitable for adjusting the observations. In most of the cases, the GEV with its special realizations of Gumbel (GbD) and Weibull (WbD) distribution appears to be appropriate (26 distributions), followed by the inverse Gaussian pdf (IGD) for five parameters and the Gamma pdf (GmD) for three parameters. Especially for flow parameters derived from the soundings, GEV appears to be the most appropriate (19 out of 28 cases). In




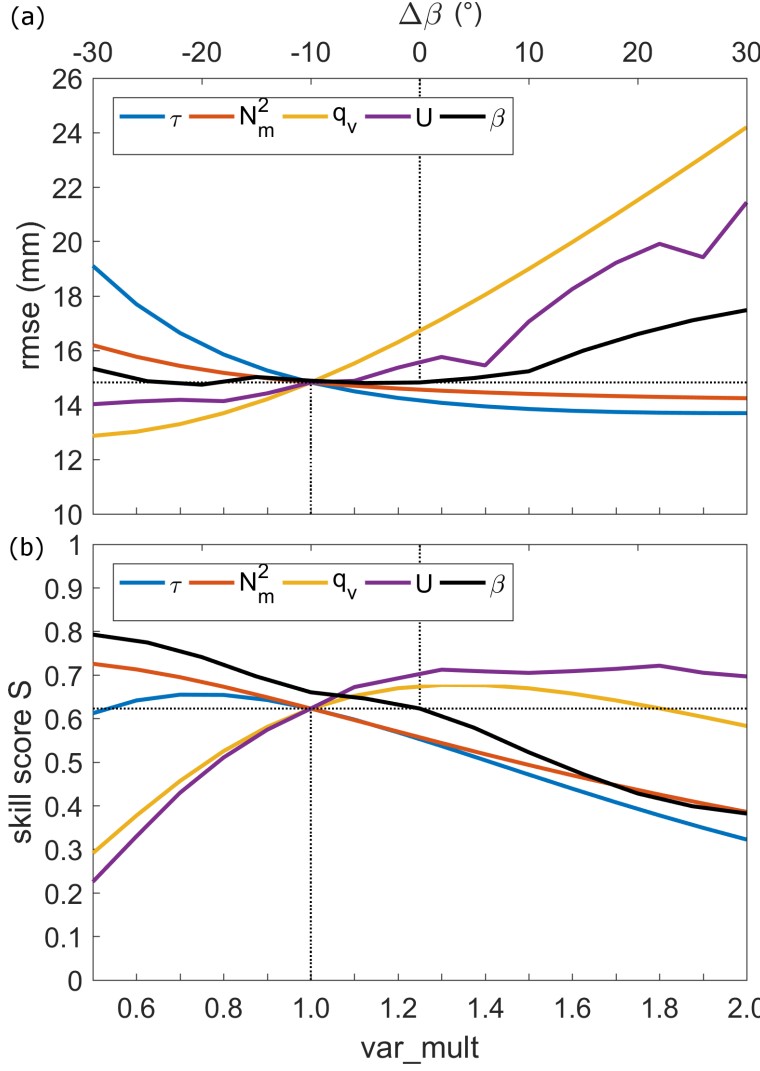

**Figure 8.** Changes of (a) rmse, and (b) skill score $S$ for perturbed values of $N_\mathrm{m}^2$, $q_\mathrm{v}$, $U$, $\beta$, and $\tau$, with a multiplicative factor ($var\_mult$), and changed $\Delta\beta$, for 31 May, 2013. The dotted lines indicate the values of the reference run.

five out of 40 cases ($\approx 12.5\%$), we had to choose the pdf manually due to the alikeness of two pdfs according to the QIRS method.

## 5.2 Event characteristics

Based on REGNIE data and the method described in Section 2.4, we estimate for each event within top200 duration $t_\mathrm{ev}$ (in

5 days), again differentiating among the seasons. The histogram of historic events and the corresponding best-fitting pdf (Fig. 9)





**Table 3.** Estimated best fitting pdfs for event duration ($t_{\mathrm{ev}}$), background precipitation $R_{\infty}$, and frontal enhancement factor $c_{\mathrm{front}}$ derived from REGNIE data (top box); square of saturated Brunt-Väisälä frequency $N_{\mathrm{m}}^2$, wind direction $\beta$, horizontal wind speed $U$, water vapor scale height $H_{\mathrm{w}}$, actual lapse rate $\gamma$, saturated moist adiabatic lapse rate $\Gamma_{\mathrm{m}}$, and condensation rate $\rho_{S_{\mathrm{ref}}}$ derived from sounding data (bottom box); for the pdf aconyms: see Table 1.

| model parameter | MAM | JJA | SON | DJF |
|:---:|:---:|:---:|:---:|:---:|
| $t_{\mathrm{ev}}$ | GEV | GEV | BSD | NkD |
| $R_{\infty}$ | WbD | WbD | WbD | WbD |
| $c_{\mathrm{front}}$ | LND | GmD | LND | ND |
| $N_{\mathrm{m}}^2$ | GEV | GbD | GEV | GEV |
| $\beta$ | GEV | GEV | GEV | SD |
| $U$ | HND | IGD | HND | GEV |
| $\gamma$ | GEV | GEV | IGD | IGD |
| $\Gamma_{\mathrm{m}}$ | GEV | IGD | IGD | GEV |
| $H_{\mathrm{w}}$ | GEV | GbD | GEV | LD |
| $\rho_{S_{\mathrm{ref}}}$ | WbD | GEV | WbD | WbD |

shows that during the summer (JJA), a duration of between two and three days dominates with a decreasing probability toward longer periods. In the winter (DJF), the distribution is generally shifted to longer events, whereas the probability for single-day events remains roughly unchanged. The maximum of 15 days in DJF represents the longest duration of top200. Whereas the estimated pdf for the summer (GEV) has a sharper maximum and a stronger decrease for $t_{\mathrm{ev}} > 3$, the pdf found to best fit the duration in the winter (NkD) shows a broader range of possible durations. Note, however, that the histogram in the winter shows a large scattering with irregular peaks, making an adjustment to a pdf very problematic. For the spring and autumn, the results are comparable to those of the winter and summer respectively.

Concerning background precipitation $R_{\infty}$, totals of 20–25 mm d$^{-1}$ are found to most likely occur with a range of 3–37 mm d$^{-1}$ in the winter, 3–50 mm d$^{-1}$ in the summer, and 0–50 mm d$^{-1}$ during the other two seasons (not shown). For all seasons, the Weibull distribution (WbD) is most appropriate. For frontal factor $c_{\mathrm{front}}$, we obtain a log-normal distribution (LND) for the spring and fall, a normal pdf (ND) for the winter, and a Gamma pdf (GmD) for the summer. All pdfs have their maximums around 0.7 to 0.8 with a range from 0.4 to 1.4 for most of the seasons (not shown). The gamma distribution in the fall has a sharp ascent and a slower descent toward higher values (maximum of around 1.6).

### 5.3 Atmospheric parameters

As described in Section 3, orographic precipitation in the SPM2D depends on the atmospheric parameters (cf. Table 3). An overview of the range of all parameters is shown as box plots in Figure 10. In most cases, the atmosphere was slightly stably stratified as represented by positive values of the squared Brunt-Väisälä frequency $N_{\mathrm{m}}^2$ affecting the wave propagation. During



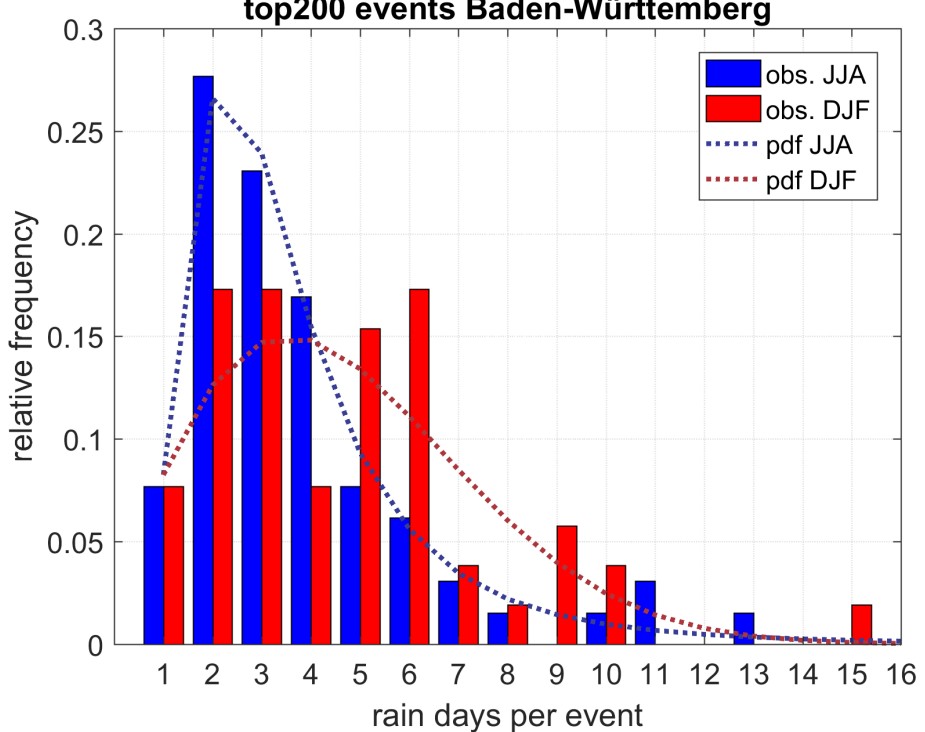

**Figure 9.** Histogram of top200 event duration for Baden-Württemberg according to REGNIE (bars), and estimated best fitting pdfs (dotted lines) for the summer (blue) and the winter (red).

the summer, the distribution is shifted toward negative values (= unstable; recall that negative values are set to $0.0003\ \mathrm{s}^{-1}$), whereas in the winter, there are almost entirely positive values. Wind direction $\beta$, decisive for the spatial distribution of precipitation around the mountains, for example, the distribution of enhanced and reduced precipitation, shows pronounced seasonal differences. More than 90% of the top200 winter events have southwesterly to northwesterly winds (240°–300°),

5   with other directions hardly observed. The reason is that northerly flows are usually associated with low temperatures and thus low humidity during the winter and do not have the potential for heavy precipitation. In the summer, the wind direction that occurred most frequently is between 240° and 300°. However, all other directions have been observed as well.

Horizontal wind speed $U$ is high, especially during the winter, where reduced moisture is compensated by high velocity to obtain substantial horizontal incoming moisture flow. Median values are 5 and $20\,\mathrm{m\,s}^{-1}$ during the summer and winter

10   respectively. Flow parameters related to humidity ($H_\mathrm{w}$, $\rho_{\mathrm{S_{ref}}}$) conversely show higher values in the summer, where $\Gamma_\mathrm{m}$ is reduced due to the release of latent heat. Observed vertical temperature gradients $\gamma$ show similar medians and interquartile ranges with a broader distribution in the winter.

In the histograms of $N_\mathrm{m}^2$ for the top200 events (Fig. 11, left), distinct seasonal differences as already discussed can be identified. For both seasons, the pdf with the best fit is a GEV (special case of Gumbel in summer), yielding a longer tail to





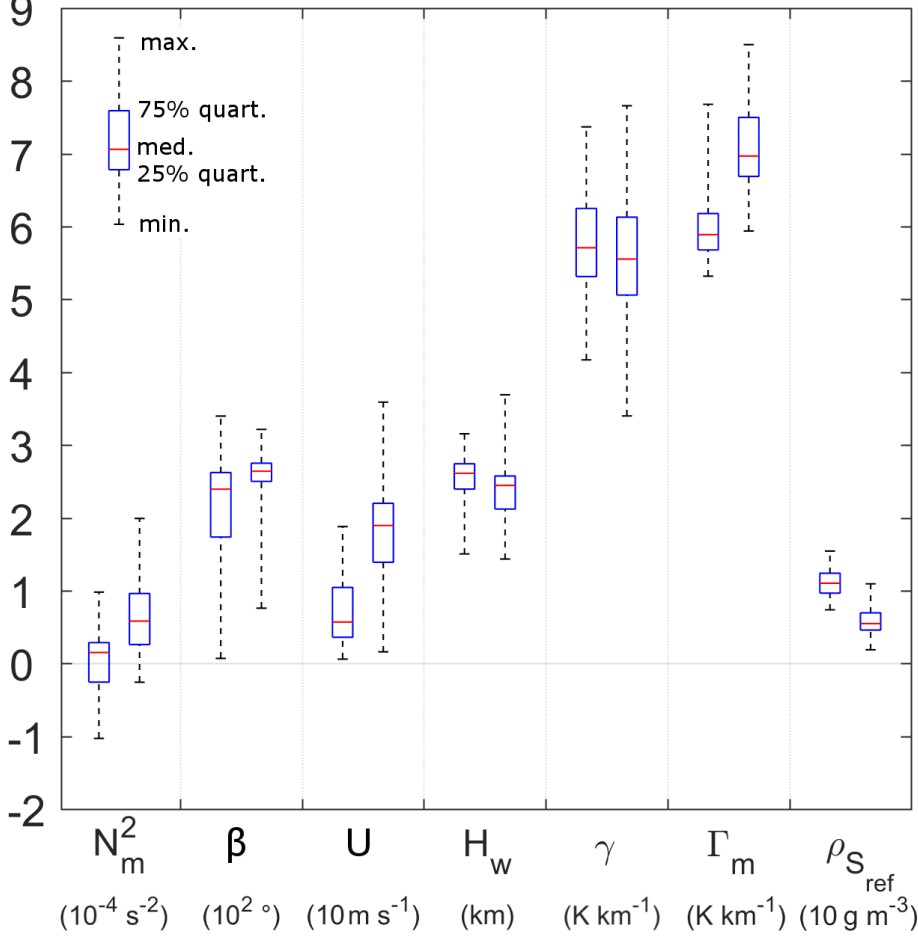

**Figure 10.** Atmospheric parameters required for the SPM2D derived from radiosounding observations at Stuttgart for top200 events with mean, interquartile distance, minimum, and maximum values; the left box-whisker of each pair represents the summer, the right one represents the winter season. The units for each variable are given in the brackets below the variable names.

the right in the winter and a longer left tail in the summer. Seasonal differences are also well pronounced for wind direction $\beta$ (Fig. 11, right). Whereas during the winter time directions outside of 210° and 320° can be neglected, the distribution during the summer shows several irregularities, making the adjustment of the distribution function difficult. However, it seems to be plausible to apply a GEV distribution in the summer and the Stable distribution (SD) in the winter.



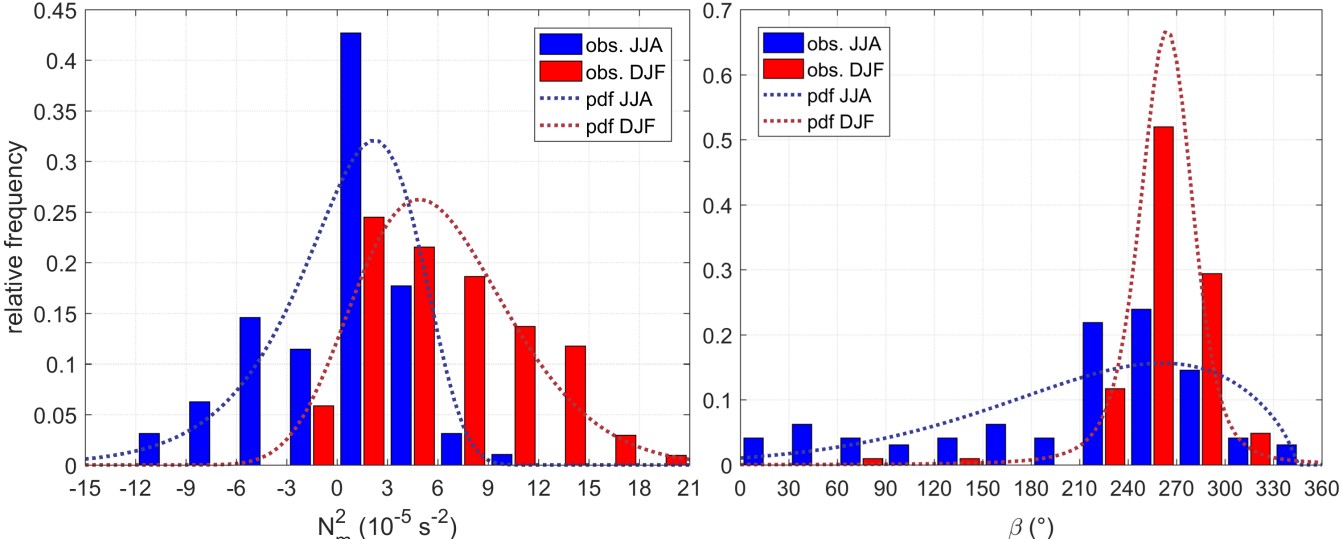

**Figure 11.** Histogram of top200 observations (bars), and estimated best fitting pdfs (dotted lines) for square of saturated Brunt-Väisälä frequency $N_m^2$ (left), and mean wind direction $\beta$ (right); the blue colors indicate the summer data (JJA), the red colors the winter data (DJF). The bars are located around the center of a histogram class.

## 6 Stochastic event set

Overall, a total number of 10,000 events (approx. 31,500 days) have been simulated with the SPM2D, hereafter referred to as the SPM10k. For the evaluation with REGNIE data, we quantified statistical values, such as return periods, probabilities, or percentiles.

Spatial 24-hour mean values for the area of Baden-Württemberg range between 1.2 mm and 79.7 mm in the SPM2D, whereas the maximum for top200 is only 49.6 mm. In total, 128 events (or 0.4%) of the SPM10k yield higher spatial precipitation amounts. Both median and 90th-percentile (p90) precipitation fields of top200 events and the SPM10k agree well concerning the spatial distribution as well as the precipitation amounts (Figure 12). Significant orographic structures in the precipitation fields over the Black Forest and Swabian Jura are clearly visible in both data sets. The areal rainfall of the SPM10k median field differs only about 3.3% from top200, whereas that of the p90 field is about 6.5% smaller. Maximum values in the SPM10k are about 7% higher for the median field and approximately 1% smaller for the p90 field. Note that the more detailed structure of REGNIE data results from the regionalization method and its strong dependency on orography and should not be over-interpreted. Larger spatial differences mainly appear in the northern parts of Baden-Württemberg (the Northern Rhine Valley and northeastern rolling hills) for both the median and the p90 field, whereas for the latter, differences also arise in an additional area northeast and southwest of Stuttgart. Nevertheless, all differences are small in the order of a few percent.

Comparing precipitation amounts for other percentiles, for example, between the 16th and 99th percentiles (Figure 13 a), the differences between REGNIE and the SPM2D are very small for the spatial mean values and the maximum precipitation at any





**Figure 12.** Precipitation fields for the median (top), and the 90th percentile (p90; bottom) of top200 (REGNIE) events (left column), and the SPM10k (right column).

grid point in the model domain. The differences become considerable only for the 95th percentile or above. The SPM2D tends to overestimate lower precipitation amounts because the minimum values at any grid point are higher in the model than in the observations and invert for the 99th percentile only. At small percentiles, or for small precipitation amounts, respectively, QIs, such as correlation coefficient $r$, skill score $S$, and normalized standard deviation $\hat{\sigma}_{\mathrm{f}}$, have low values due to the overestimation of the SPM2D (Figure 13 b). The highest skill is reached around the 90th percentile with a slight decrease for higher values, which can be the result of the increasing uncertainties of the observations. Nevertheless, a skill score of around or above 0.8 confirms the reliability of the simulations.

To estimate precipitation distributions for specific return periods, we fit a Gumbel distribution (Wilks, 2006) to the annual maximum series of both REGNIE and the SPM10k. Because it is not possible to estimate the time period and a corresponding





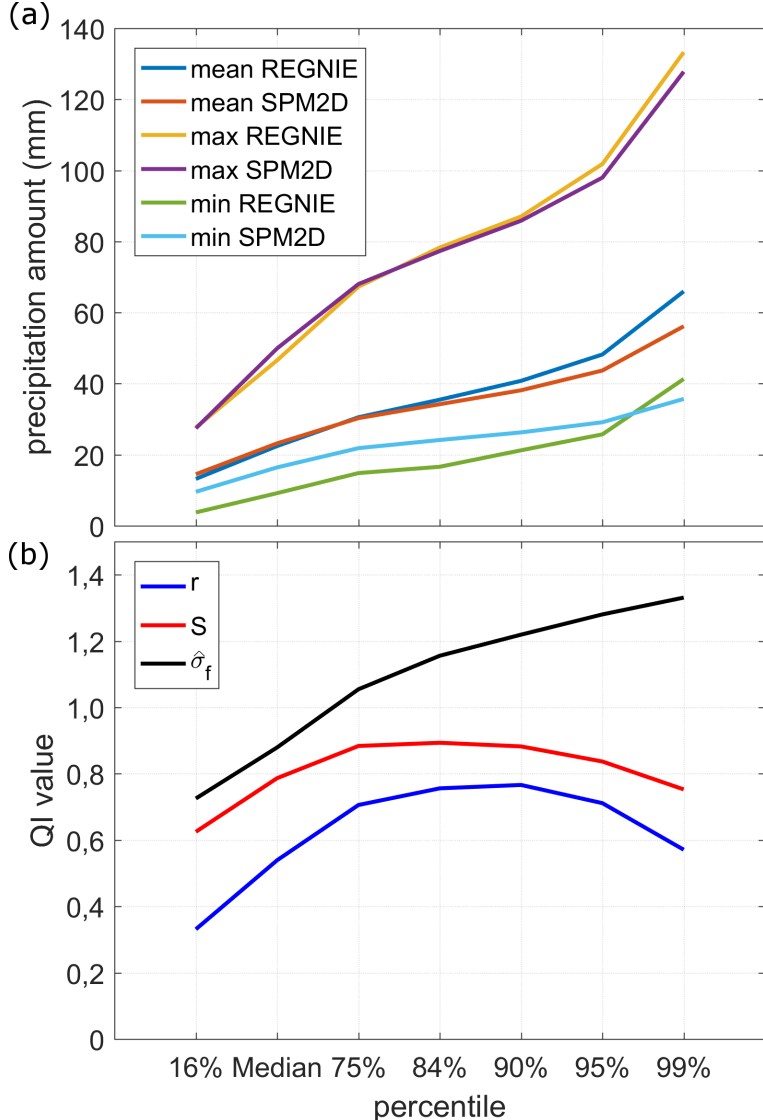

**Figure 13.** Comparison of (a) the maximum, the minimum, and the spatial mean precipitation of REGNIE and the SPM2D, and (b) quality indices (QI) $r$, $S$, and $\hat{\sigma}_\mathrm{f}$ for different percentiles.

annual maximum series for the stochastic event set, we count the number of stochastic values exceeding the 99th percentile of observations $n_\mathrm{p99}$ and normalize it by the probability of occurrence $p_{99}$, producing new time period $T_\mathrm{SPM}$:

$$T_\mathrm{SPM} = \frac{n_\mathrm{p99}}{p_{99}} \tag{13}$$

After sorting the SPM10k in descending order, we take the first $n_T = T_\mathrm{SPM}$ values as the annual series of the SPM10k and

5  estimate a Gumbel distribution. Using the distribution parameters, we obtain precipitation values for specific return periods for





both the observations and the SPM10k. This method is applied to the spatial mean values of different areas and for every single grid point.

For a 10-year return period, the SPM10k shows only small deviations from REGNIE of less than $\pm 10\%$ over almost the entire area of Baden-Württemberg, with a small area of overestimation in the Southern Black Forest (Figure 14 a). The areal

mean difference is only 0.6%. In the case of $T = 200$ yrs (Figure 14 b), the overestimation in the Southern Black Forest remains with almost the same relative discrepancy. For this return period, the SPM10k tends to underestimate precipitation, especially in the northern part of Baden-Württemberg and in the southeast around Lake Constance. Nevertheless, the deviations for most of the grid points are between $\pm 20\%$, and the areal mean difference is about –10%. Taking into account the strongly increasing uncertainties of the observed values for higher return periods, especially for $T > 100$ years, this is still a reasonable result.

On the level of the major river catchments, the differences also are small. For the Neckar catchment, for example (see Figure 14), which covers about 38% of Baden-Württemberg, the spatial mean deviation is about –0.5% in the case of $T = 10$ yrs and –12.7% for the 200-year return period. Even for the catchments containing the area of overestimation in the Southern Black Forest (Upper Rhine between Basel and Mannheim, and High Rhine between Constance and Basel), the spatial mean deviations are between +1 and +4% for $T = 10$ yrs and between –2 and –10% for $T = 200$ yrs respectively.

Single grid point deviations and the ensuing spatial mean values as described above are sensitive to local conditions and uncertainties in both REGNIE and SPM10k data. Hence, we evaluate the model in a similar way by calculating the spatial mean precipitation first and then fitting a Gumbel distribution to the spatial means in a second step. For the plotting, return period $T_k$ of each element $x_k$ of the annual maximum series with length $T_{max}$ is given by $T_k = T_{max} \cdot \mathrm{rk}(x_k)$ with the rank of element $x_k$ $\mathrm{rk}(x_k)$ (annual series sorted in descending order). The first element (highest value) of an annual series of, for

example, 100 years therefore has a return period of $T_1 = 100$ yrs, the second $T_2 = 50$ yrs, and so on. The values of $T_k$ were adjusted using the plotting position method of Cunnane (1978).

Again, the difference between the simulated and observed spatial mean values of daily precipitation for the whole of Baden-Württemberg is small, with slightly lower values from the simulations (Figure 15 a). The distribution of the SPM10k is very close and almost parallel to the estimated observed Gumbel distribution and mostly in between or close to the 95% confidence

interval (CI95) estimated with the formula of Dyck (1980). Considerable differences between the SPM10k and REGNIE arise only for return periods of $T = 1000$ yrs and above but are still small. For the Neckar catchment, the simulation results agree well with the observed distribution for return periods up to approximately 300 years (Figure 15 b). For higher return periods, the differences increase but are still inside or around the CI95. Similar results can be found for other river catchments. Note again that for such high return values, the statistical uncertainty of the observed distribution also increases significantly.

# 7   Summary and Conclusions

We have presented a novel method for estimating the statistics of total rainfall based on a stochastic model approach (SPM2D). Total precipitation at any grid point is calculated from the linear superposition of four different parts: orographic precipitation, synoptic background precipitation, frontal precipitation and embedded convection. The linear theory of orographic precipita-





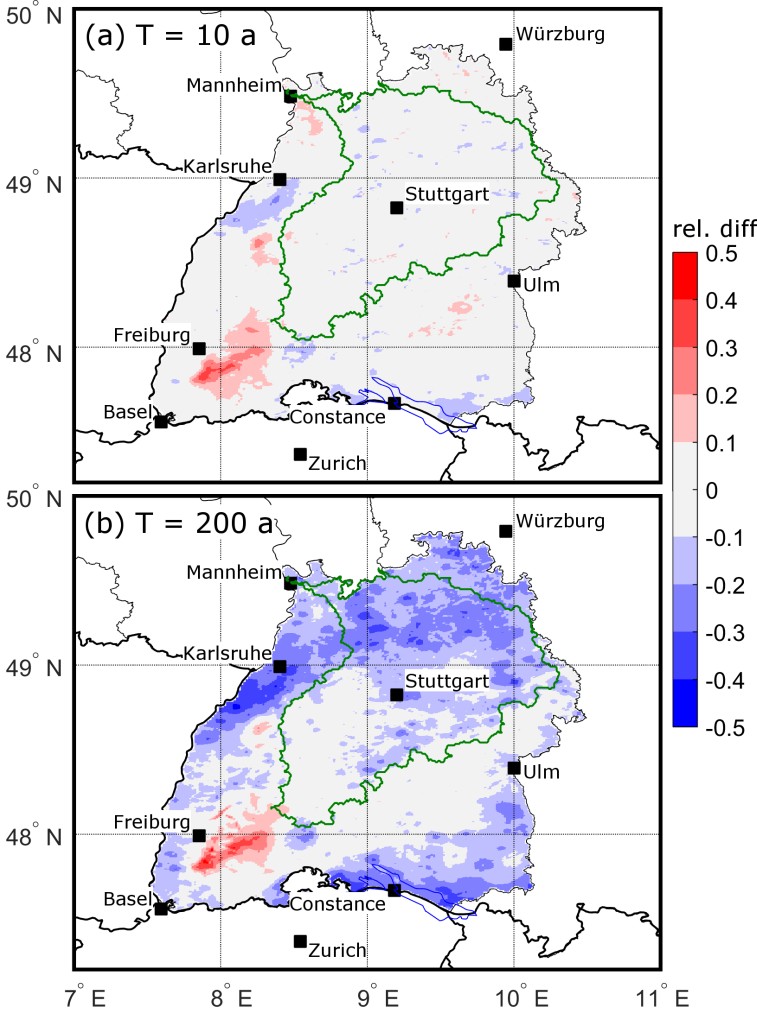

**Figure 14.** Relative difference of the precipitation amounts for (a) a return period of $T = 10$ years, and (b) $T = 200$ years, according to a Gumbel distribution fitted to the observations (top200) and the SPM10k (see text for further explanation). The Neckar catchment is shown as green contour.

tion according to Smith and Barstad (2004) has been modified by calibration parameters to minimize the weaknesses found in previous studies (e. g., Barstad and Smith, 2005; Kunz, 2011) and to adjust the model to the specific conditions of the investigation area. We calibrated and adjusted the SPM2D to a historic event set of heavy rainfall events (top200). Using probability density functions (pdfs) for all required model parameters, we simulated 10,000 stochastic precipitation events and compared

5  the results with observations using different percentiles or return periods.

The focus of the presented investigations was on the Federal State of Baden-Württemberg in Southwest Germany. The following main conclusions can be drawn:





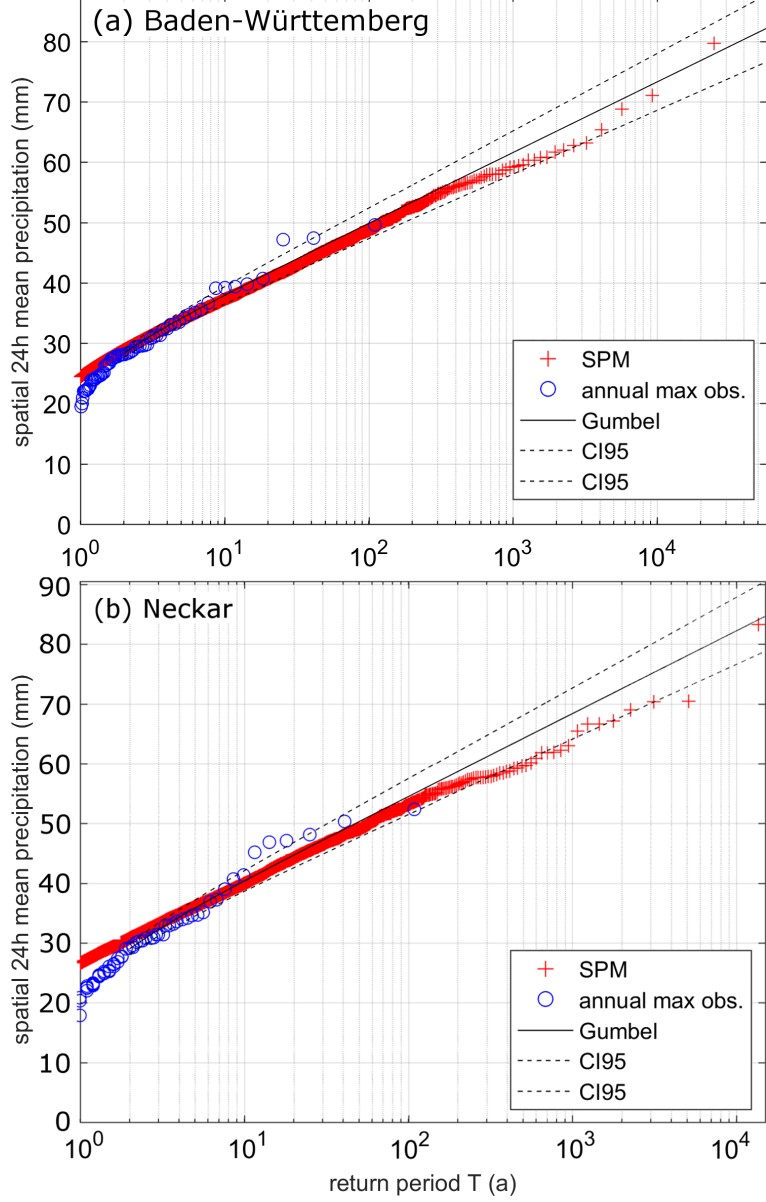

**Figure 15.** Daily rainfall totals (areal mean) as a function of return period $T$ based on the annual maximum series of observations (REGNIE, blue), the corresponding Gumbel distribution including the 95% confidence intervals (black), and the annual SPM10k series (red) for (a) the Federal State of Baden-Württemberg, and (b) the Neckar catchment.

- The results illustrate the capability of the SPM2D to both simulate historic and stochastic events with realistic spatial distribution and magnitude despite its simple approach of using just seven atmospheric variables for precipitation calculation. The differences between the SPM2D and REGNIE are small with deviations of less than 10%. Local differences





may be the result of an inhomogeneous distribution of rain gauges, which leads to discrepancies in the interpolated REGNIE fields. The interpolation method used for REGNIE furthermore overemphasizes the orographic influence on the precipitation distribution whereby more uncertainties emerge.

- The solution of the model equations in the Fourier space using a fast Fourier transform algorithm allows for simulations of a large number of events with less temporal effort. Furthermore, wave dynamics are directly implemented.

- The linear approach for orographic precipitation worked quite well during calibration to historic events (top200). The newly added amounts for frontal precipitation and embedded convection as well as the implemented model parameters afford even more plausible realizations on a physical basis: $f_{C_w}$ decreases the sensitivity of orographic precipitation on multiple ascents; $f_{dry}$ reduces the underestimation of lee-side precipitation due to evaporation; and $c_{oro}$ takes into account the assumption of an entirely saturated atmosphere at any time.

- The presented stochastic approach is easily applicable to other investigation areas. Prerequisite information of atmospheric variables can be estimated based on radiosoundings, as within this study, or based on reanalysis or forecast data of numerical weather or climate models. Therefore, it can be used for precipitation simulations in areas with less or even no ground-based observations.

As shown in a case study, the SPM2D is sensitive to perturbations of ambient conditions, and therefore, high-quality input data, especially of the atmospheric parameters, are essential. Both radiosoundings or numerical model outputs may not correctly represent the undisturbed conditions upstream of the investigation area due to, for example, an inconvenient location of the launching station of the radiosondes or measurement errors, which leads to hardly quantifiable uncertainties in the SPM2D.

The SPM2D is part of a risk assessment methodology that estimates the flood risk for a local direct insurer. Within the framework of this project, the SPM2D is used for two more federal states in Central Germany with the quality of the results resembling those presented for Baden-Württemberg in this study. The precipitation fields simulated with the SPM2D are used as input data for hydrological simulations with the aim of the flood risk estimation of a one-in-200-year event and the corresponding probable maximum loss (PML200), as it is mandatory by law for the insurance industry to maintain capital stock for such an event.

Besides the PML200 for insurance companies, the proper statistics of extreme events are useful for multifarious issues in different fields of water management, such as flood protection with dams or retention areas, or forecast scenarios.

## 8 Data availability

The REGNIE data used in this paper are freely available for research and can be requested at the DWD (doi:10.1127/0941-2948/2013/0436); The sounding data are freely available from the Integrated Global Radiosonde Archive (https://www.ncdc.noaa.gov/data-access/weather-balloon/integrated-global-radiosonde-archive). The required orographic data set (doi:10.1080/13658810601169899) can be downloaded from http://srtm.csi.cgiar.org/.

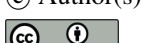



*Competing interests.* The authors declare that they have no conflict of interest.

*Acknowledgements.* The authors thank a local insurance company for funding the project. We also would like to thank the German Weather Service (DWD) and the Integrated Global Radiosonde Archive (IGRA) for providing different observational data sets and CGIAR-CSI for the orographic data. Special thanks go to James Daniell, Andreas Kron and Simon Hoellering from KIT for constructive discussions within
5   the project and for valuable suggestions during the model development. We acknowledge support by Deutsche Forschungsgemeinschaft (DFG) and open access publishing fund of Karlsruhe Institute of Technology (KIT).





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
