# Peer review of "Flood-Related Extreme Precipitation in Southwestern Germany: Development of a Two-Dimensional Stochastic Precipitation Model"

_Hydrology and Earth System Sciences, 2018_

## Referee Comment (RC1) · Anonymous Referee #1 · 24 May 2018

This study extended the previous models from Smith and Barstad (2004) and Barstad and Smith (2005) to stochastically generate extreme precipitation events. The model relates extreme precipitation to atmospheric conditions, kind of circulation-based model. This model is exclusively for extreme precipitation events, different from those models for long-term weather generation. The paper presented a lot of details to interpret the procedures about development, calibration and validation of the proposed model. The topic falls within the scope of HESS.

Although the manuscript gave enough information about the model, it is not so easy to

follow in the current form. I strongly suggest adjustments of the paper structure. First, a flowchart should be given to show the development, calibration and validation of the model. Second, it is better to first give the model description following by data description, which is the usual way for method development. Third, it is necessary to simplify some sections, but focus on how to connect atmospheric conditions with extreme precipitation so that the modeled data can represent the regional condition instead of one site. Fourth, usually, for model development, a comparison with a paralleled model is necessary. Please consider the possibility to add this part. Although it takes time to do additional comparison, it is persuasive to highlight the strength of your model. Further, people would wonder how your model's performance compare with the models for long-term weather generation. With the above adjustments, the manuscript would be easier for readers to understand.

Furthermore, the authors should state the potential extension of the proposed models to the other regions in the world, which would be helpful for readers to know how to use it. Otherwise, it is a model just applicable to a specific region, which is not necessary to publish it in an international journal.

---

## Referee Comment (RC2) · Anonymous Referee #2 · 29 May 2018

This is a difficult paper to read – I think some rewriting and tightening would help exposition and the reader understand what the main contributions are. In terms of the scientific problems, my major concerns can be summarized by the following comments:

1) How can you be sure the model isn't overfit? There are numerous parameters and features, and a seemingly exhaustive parameter estimation method is used, but shouldn't there be a cross-validation study where training data are used to fit the model, and held-out testing data are used to validate the goodness-of-fit?

[Figure]

2) Lack of comparison against a simpler model. There are many moving pieces in this model; which components are giving the most improvement? In particular, it would be helpful to consider simpler versions of the model and compare their relative performance in simulation, this would help the reader understand which contributions are the most important and where future research may focus.

3) How well are spatial correlations maintained in the model? Spatially aggregated statistics like max, min and means are validated, but what about raw correlations?

---

## Referee Comment (RC3) · Anonymous Referee #3 · 30 May 2018

The authors have taken a physically-based, simplified model of orographic precipitation and added mitigations in their approach. The approach has been tested with good results.

It is an interesting and valuable contribution to the literature on this subject. It is thoroughly done and, given the complexity of the approach, it is easy to follow. I would say the results are convincing and robust. Below is a few comments/questions.

Main question: It seems to me that the input parameters are treated independently,

section 5.1, in this approach. We know that input parameters such as wind speed and direction are not independent, and thus should not be threated as such. Categorization helps, but still leaves us with the problem mentioned above. If I have understood this correctly, how do you justify independence (picking from pdf's in a random fashion)?

Minor comments/questions:

P7, L12, " linear model assumes penetration through the whole atmosphere...": Does it? it is contrary to what you write below Eq. 6, L24 which I thought was the idea of wave dynamics; reduced penetration with height. Perhaps the over-estimation has something to do with the saturation assumption you mentioned?

P9: If c_oro is constant in the whole domain, it could be enter in wave space. Can it be collapsed with f_Cw into a common factor, reducing the number of free parameters?

Fig 15: I believe that the confidence interval should be wider on the upper side than the lower side (due to fewer data points).

---

## Author Comment (AC1) · 9 Jul 2018

**Point-by-point response to Reviewer #3**

Florian Ehmele on behalf of the co-author

July 09, 2018

Thank you very much for your work and the useful and valuable comments that helped to improve the scientific quality of our manuscript. Please find below our reply to the individual points.

*The authors have taken a physically-based, simplified model of orographic precipitation and added mitigations in their approach. The approach has been tested with good results.*

*It is an interesting and valuable contribution to the literature on this subject. It is thoroughly done and, given the complexity of the approach, it is easy to follow. I would say the results are convincing and robust. Below is a few comments/questions.*

*Main question: It seems to me that the input parameters are treated independently, section 5.1, in this approach. We know that input parameters such as wind speed and direction are not independent, and thus should not be threated as such. Categorization helps, but still leaves us with the problem mentioned above. If I have understood this correctly, how do you justify independence (picking from pdf's in a random fashion)?*

This is a very helpful comment. To address this point, we will add a new section "4.3 Model sensitivities" to the paper with a more detailed sensitivity study of total precipitation to varying initial conditions including discussion in Section 5.1. Our results show different behaviors of the correlations between the input variables. Overall, the relation between the input parameters is weak with correlation coefficients in most cases between +/- 0.3, and only for two parameters of +/- 0.7. After seasonal differentiation, there are significant correlations in only one season. Those cases with higher correlation are mainly related to stability (saturated Brunt-Väisälä frequency $N\_m^2$). As shown in Figures 7 and 9, however, the model is less sensitive to this parameter compared to other. Taking into account the three points mentioned above, we found it acceptable to treat the input variables independently to keep the SPM as simple as possible. We will add a statement on this.

*Minor comments/questions:*

*P7, L12, " linear model assumes penetration through the whole atmosphere...": Does it? it is contrary to what you write below Eq. 6, L24 which I thought was the idea of wave dynamics; reduced penetration with height. Perhaps the over-estimation has something to do with the saturation assumption you mentioned?*

This was incorrect as wave dynamics show vertically tilted waves that also decay with height (expect when Fr = U/NH is very large and where the solution more or less resembles the simple upslope approach). Furthermore, you are right that the assumption of saturation over all atmospheric layers, where also the lifting condensation level is at the surface, may lead to an overestimation of modelled precipitation. We will correct/change this in the text.

*P9: If c_oro is constant in the whole domain, it could be enter in wave space. Can it be collapsed with f_Cw into a common factor, reducing the number of free parameters?*

This might be possible. However, these two parameters affect different physical processes. f_Cw acts to reduce the uplift sensitivity of the model; therefore, it mostly affects areas with strong gradients in orography (compare Figure 2), whereas over less gradients with less orographic lifting the effect is weak. Additionally, multiple ascends/descents are possible without changes in water vapor content of the air parcel. Even though c_oro has the same effect, this parameter is independent from any lifting process and is applied throughout the domain. As mentioned in the text, it is a consequence of the assumption that vertical lifting of the entire column of air leads to condensation and instantaneous fallout of hydrometeors at any time. To deal with the resulting overestimation of available precipitable water, c_oro was implemented. We will change the text to better understand this point.

*Fig 15: I believe that the confidence interval should be wider on the upper side than the lower side (due to fewer data points).*

Yes, you are right. After checking the data and the routine, we conclude that the used empirical formula from Dyck (1980) is not the proper way. We redid the plots of Figure 15 (old numbering) using the statistical calculation described by Maity (2018) and replaced it in the new manuscript version.

**References**

Dyck, S., 1980: Angewandte Hydrologie, Teil 1: Berechnung und Regelung des Durchflusses der Flüsse., 2 edn.

Maity, R., 2018: Statistical Methods in Hydrology and Hydroclimatology, Springer Nature Singapore Pte Ltd., https://doi.org/10.1007/978-981-10-8779-0, 2018.

---

## Author Comment (AC2) · 9 Jul 2018

**Point-by-point response to Reviewer #2**

Florian Ehmele on behalf of the co-author

July 09, 2018

Thank you very much for your work and the useful and valuable comments that helped to improve the scientific quality of our manuscript. Please find below our reply to the individual points.

*This is a difficult paper to read – I think some rewriting and tightening would help exposition and the reader understand what the main contributions are.*

We understand the reviewer's point that the manuscript may be difficult to follow. In the revised version of the manuscript, we try to improve the readability by re-organizing the sections and by tightening / deleting details that are not that important. We will also add a flow chart showing the different components of the model as suggested by Reviewer #1 to better understand the model's different components and their links.

*In terms of the scientific problems, my major concerns can be summarized by the following comments:*

*1) How can you be sure the model isn't overfit? There are numerous parameters and features, and a seemingly exhaustive parameter estimation method is used, but shouldn't there be a cross-validation study where training data are used to fit the model, and held-out testing data are used to validate the goodness-of-fit?*

We don't really see a conflict with overfit or "overengineering". All four components of the model (orographic, background, frontal, embedded-convection) are physically-based processes related to vertical lifting on different scales. The parameter estimation mainly is required for the dynamical core (wave dynamics) and the microphysics. For clarification, we will add a comment on this in the conclusion section. Considering the question of cross-validation, we fully agree. However, this was already done. The model is trained for a set of approx. 100 events and then driven stochastically over 10.000 events. The stochastic simulations, the main purpose of the model, are evaluated against the observations using different statistical quantities. To avoid any confusion, we will highlight this in the manuscript.

*2) Lack of comparison against a simpler model. There are many moving pieces in this model; which components are giving the most improvement? In particular, it would be helpful to consider simpler versions of the model and compare their relative performance in simulation, this would help the reader understand which contributions are the most important and where future research may focus.*

We agree that adding a comparison to simpler versions or to other models would highlight the potential and skill of our approach (this point was also recommended by reviewer #1). As to the best of our knowledge there is no comparable stochastic model available, we will use COSMO-CLM reanalysis instead, but focusing on historic events. We will split the four-part Figure 12 into two Figures (new 13 and 14), one for the median and one for the $90^{th}$ percentile and add the corresponding statistics for both cases using the reduced stochastic model (rSPM, basic model) and reanalysis data performed with the COSMO-CLM model. We same will apply for Figure 13 (new Fig. 15). With those additions the improvements of the model should become clearer.

*3) How well are spatial correlations maintained in the model? Spatially aggregated statistics like max, min and means are validated, but what about raw correlations?*

We are not sure what the reviewer exactly means with "raw correlations". The presented analysis of maximum and minimum values in Figure 13 is not spatially aggregated, but show max/min values at each grid point in the model domain. The median and percentile values presented in Figure 12 as well as the difference between model and observations for different return periods shown in Figure 14 are grid point-based statistics and not spatially aggregated. As the SPM is event-based and not designed for continuous simulations of extreme events it is not possible to correlate "time series" between simulations and observations for the different grid points. Considering wave dynamics in combination with an FFT algorithm, a spatial conjunction is already given.

---

## Author Comment (AC3) · 9 Jul 2018

**Point-by-point response to Reviewer #1**

Florian Ehmele on behalf of the co-author

July 09, 2018

Thank you very much for your work and the useful and valuable comments that helped to improve the scientific quality of our manuscript. Please find below our reply to the individual points.

*This study extended the previous models from Smith and Barstad (2004) and Barstad and Smith (2005) to stochastically generate extreme precipitation events. The model relates extreme precipitation to atmospheric conditions, kind of circulation-based model. This model is exclusively for extreme precipitation events, different from those models for long-term weather generation. The paper presented a lot of details to interpret the procedures about development, calibration and validation of the proposed model. The topic falls within the scope of HESS.*

*Although the manuscript gave enough information about the model, it is not so easy to follow in the current form. I strongly suggest adjustments of the paper structure. First, a flowchart should be given to show the development, calibration and validation of the model.*

We understand the reviewer's point that the manuscript may be difficult to follow. In the revised version of the manuscript, we try to improve the readability by re-organizing the sections and by tightening / deleting details that are not that important. We follow the suggestion and included a flow chart with corresponding descriptions in the text that may help to better understand the links among the different components.

*Second, it is better to first give the model description following by data description, which is the usual way for method development.*

In the new version of the manuscript, we changed the order of Sections 2 and 3 as suggested. We also rearrange some Sections to improve the logical story line.

*Third, it is necessary to simplify some sections, but focus on how to connect atmospheric conditions with extreme precipitation so that the modeled data can represent the regional condition instead of one site.*

Ambient conditions directly feedback into the model equations that combine flow conditions with microphysics, thus representing the regional conditions. This is highlighted, for example, in Figures 7 and 9 (which will be Figs. 8 and 10 in the revised version). Furthermore, to highlight directly the relation between environment and ambient conditions, we will include a new Figure. Another critical point is that we used only data from one radiosounding (Stuttgart). As shown by Kunz (2011) for a comparison between Stuttgart and Nancy sounding, ambient conditions during large-scale heavy rainfall usually do not show large gradients (at least for the parameters considered in the model and without fronts that are, thus, treated separately). We will add a comment in the manuscript.

*Fourth, usually, for model development, a comparison with a paralleled model is necessary. Please consider the possibility to add this part. Although it takes time to do additional comparison, it is persuasive to highlight the strength of your model. Further, people would wonder how your model's performance compare with the*

*models for long-term weather generation. With the above adjustments, the manuscript would be easier for readers to understand.*

We agree with the reviewer that a comparison with other models would be appropriate to highlight the skill and characteristics of our model. However, we are not aware of any comparable large-scale two-dimensional stochastic precipitation model. Therefore, we will compare the full SPM2D using the basic setup (reduced SPM; rSPM) with COSMO-CLM (CCLM) reanalysis using the top200 events. For this, we split Figure 12 into two new Figures 13 and 14, one for the median and one for the 90[th] percentile, and add the corresponding statistics of the rSPM and CCLM simulations. The same will apply to Figure 13 (new Fig. 15).

*Furthermore, the authors should state the potential extension of the proposed models to the other regions in the world, which would be helpful for readers to know how to use it. Otherwise, it is a model just applicable to a specific region, which is not necessary to publish it in an international journal.*

The methodology is not limited to a specific region. The basic core of the model, the orographic rainfall model according to Smith and Barstadt (2004), has been applied successfully to several regions around the world (e.g., US, Norway, Iceland, Germany). Our extension, the stochastic approach, only requires precipitation totals to estimate background and frontal precipitation including calibration. We will add a comment about the potential transferability in the conclusion section.

**References**

**Kunz, M.:** Characterisitics of Large-Scale Orographic Precipitation in a Linear Perspective, *J. Hydrometeorol.,* **12**, 27–44, 2011.

**Smith, R. B. and Barstad, I.:** A Linear Theory of Orographic Precipitation, *J. Atmos. Sci.,* **61,** 1377–1391, 2004.

---

## Editor Comment (EC1) · J. Seibert (Editor) · 10 Jul 2018

The reviewers raise a number of important points. Besides some need for clarifications, a major issue is that the new approach should be compared to some alternative model. Here the responses of the authors are not fully satisfactory. I would argue that it, with some creative thinking, should be possible to come up with some suitable benchmarks.

---

## Referee Report (RR1)

Dear Florian Ehmele and Michael Kunz,

I have now finished reading your manuscript entitled "Flood-Related Extreme Precipitation in Southwestern Germany: Development of a Two-Dimensional Stochastic Precipitation Model". I found the manuscript interesting and I think the hydrological community will benefit from the advancements in Smith and Barstad's stochastic rainfall model you are suggesting. The manuscript fall within the topics cover in HESS, and I believe that after revising the text the paper can be accepted for publication. Saying that, I have struggled to read the paper till the end. In its present form the text is hard to read and to follow. My main comments are therefore related to the structure and length of the text:

- The text is too long. I suggest moving some of the Tables and Figures to the supplementary material (SI) – see some suggestions in the specific comments below. The same goes to the text itself – some of the descriptions can be summarized in a Table (list of model parameters, for example) and some can be moved to the SI. In addition, the text can be shorten in many places. For example, Section 3 (data sets) could be summarized in a single page (2.5 pages currently). The first sub-section describing the rainfall product can be summarized in two sentences, refereeing to REGNIE and indicating the product limitations.
- There are many replicas in the text that need to be eliminated. For example, Section 2.6 "Stochastic modeling of precipitation events with SPM2D requires the adjustment of appropriate probability density functions (pdfs) to all input parameters…" and in the beginning of Section 5.1 "Stochastic model simulations are based on pdfs that are adjusted to the required parameter…". The text should be concise as possible, especially when the text is of technical nature like in this manuscript.
- Some restructuring is needed. In many cases I thought that information is missing and that the text is incomplete just to realize that the information I looked for is written later in the text. For example, in page 5 line 15 you mention that "R may become negative" and only later (page 6 line 25) you explain that even if it is negative you consider it to be zero. Another example, the model is briefly described at the introduction and then again in Section 2.1. I suggest to follow a simple structure: general introduction, description of the model component, model calibration (general calibration procedure, not tailored to the case study, so readers can understand what input is required and how to calibrate the model), short description of the study area, short description of the data sets, calibration and results of the case study, and conclusions.

Below please find my specific comments to the text. I hope that you will find my comments useful and take my criticism positively. I value your study and I hope they will assist in improving the manuscript.

Best regards,

Nadav

Specific comments

[page line]

[2 1] Actually, IDF curves can be estimated over large areas using gridded information from remote sensing. In recent years there were several advances in this direction. Search, for example, for recent publications by Francesco Marra.

[2 4-6] For more recent studies were rainfall generators were used to compute spatially distributed IDF curves I can suggest the Authors to also consider two recent papers that we (Peleg et al., 2017 and 2018) have published that address this topic explicitly.

Peleg et al. (2017). Partitioning the impacts of spatial and climatological rainfall variability in urban drainage modeling. Hydrology and Earth System Sciences, 21(3), pp.1559-1572.

Peleg, N. et al. (2018). Spatial variability of extreme rainfall at radar subpixel scale. Journal of Hydrology, 556, pp.922-933.

[2 14-25] This paragraph belongs to the method section, where the model is described.

[4 3] Why only stratiform clouds? I see in eq. (1) that the convection component is also consider. I suspect that Eq 1 is not referring to the original model but to the developments you added later. Maybe first describe simply the original model as it, and then added a section explaining the changes you are suggesting.

[4 7] "large scale lifting". What do you mean by that? upward lifting (i.e. omega component)? Isn't that part of vertical convection?

[4 9] "rSPM". If I understand right, you are modifying the original SPM2D model to be a reduced complexity model for some components while adding additional components that were not part of the original model. The fact that you have two names for the same model (SPM2D and rSPM) is confusing. Mention the changes made to the model (can also be given in a form of a table - original model components and the new model components) so it will be clear what is what.

[4 9] "we included two additional precipitation components". Which are not part of the original Smith and Barstad model? If this is the case, then equation 1 should be with only the components originally used in SPM2D and a new equation 2 is to be given with the components of rSPM. That will also resolve my above comment related to the stratiform cloud.

[4 11-13] The sentence is not that clear, please rephrase.

[general comment] Please add a table that summarize the parameters of the model.

[4 15] "is simulated". Simulated how? For each year the total numbers of events and the length for each events are sampled from a given distribution? If so, which distribution is used? Please give more details here.

[4 18-19] I am missing some information here. What is the spatial and temporal resolution of the model? Is the model daily or sub-daily?

[4 30] "sounding data". Can also be estimated from other sources, such as reanalysis data? A matter for a latter discussion.

[Figure 2] What are all the larger/smaller values from zero refer to? For example, for Roro, which arrow is for values larger than zero and which arrow to follow when values are smaller than zero? Rtot will always be larger than zero as only wet events are consider, no?

[5 9] I can understand why the time sacles are constant in time, but why they are constant in space?

[5 15] And is consider negative when embedded in equation 1 or is set to zero?

[6 24-26] OK, that answer my previous question. This should have been mentioned earlier.

[7 4] I do not see it in equation 6. Did you meant eq. 7?

[7 11] Eq. 7. Equation 6 is still on the k,l plane.

[7 15] "whereas Coro is constant over the whole domain". From Fig. 3 it seems that Coro is changing along the x-y plane and is not constant over the whole domain. Is it constant in time? I guesses you meant here to say that Coro is affecting all grid cells in all times.

[8 2-4] This explanation should be moved up, after equation 1 is introduced.

[8 30] The value of Cfront is determined from a distribution? Or is it constant for each season? Depend on the wind direction? Please add information on how the values of Cfront are defined.

[9 10] "Note, however, that the model is not foreseen to simulate purely convection". But, often the most extreme rainfall is (almost) purely convective in nature - isn't that contradict the purpose of this study, to explicitly account for the extreme (200-y return period) rainfall events?

[10 9] "convective cells". It makes sense to follow the cell structure only if the temporal resolution of the model is hourly or sub-hourly (is it? reading so far I couldn't find the information about the temporal resolution of the model), with coarser resolution you cannot detect cells' structure anymore.

[10 12] I get from this that the grid cell size is 1 km? That information should be written somewhere.

[11 3] "the spatial distribution of Cconv randomly varies between the given limits". Why? Anyhow you do not try to capture the spatial structure of the cells, so why adding another level of complexity?

[12 2] March-April-May, MAM from hereon. Same for the other seasons.

[Table 1] Instead of list of distributions that were examined, I suggest having a table summarizing the distribution used for each of the variables.

- Further reading, I see that this information is given in Table 3. I thus do not see the need in having Table 1 (maybe as supplementary material) and suggest to remove it.

[16 6] r refer here to the spatial correlation of a given rain field?

[Figure 7] Can be moved to the supplementary material.

[Figure 9] can be moved to the SI as well.

[21 7-11] There is more than the mean areal rainfall that needs to be consider when comparing maps of extreme rainfall. For example, how well the structure (dimension and location) of the heavy rainfall (e.g. above 50 mm per day) are reproduced?

[26 7] Simulated how? I guess that for a given event the parameters were sampled from the relevant distributions independently. Are the model variables (e.g. the lapse rates) corss-correlated and, if so, is the cross-correlation accounted for in the model (for example, by sampling the parameters from the pdfs using copulas). Please provide more information here.

[new table] I suggest adding a new table summarizing the different models and experiments (rSPM10k, CCLM, rSPM…). It becomes difficult to follow all the names.

[Figure 16] SI

---

## Author Response (AR2)

**List of Major Changes**

**Global changes:**

- Revision of text structure according to readability and story line
- Supplementary Material: Some Figures are put to a supplementary document
- We added an appendix section and moved some information into it

**Specific changes:**

- We rearranged Section 2 for a better story line and readability
- We adjusted some Figures according to the comments of the reviewer
- We reduced the number of acronyms to reduce confusion

**Point-by-point response to Reviewer #1**

Florian Ehmele on behalf of the co-author

December 19, 2018

Thank you very much for your work and the useful and valuable comments that helped to improve the scientific quality of our manuscript. Please find below our reply to the individual points.

*The authors have made great efforts in improving the manuscript. The paper structure looks better, and it appears to be a complete model development study after adding a comparison with another model.*

We thank you for your positive feedback.

*But I think the description of the model algorithm should be simplified, especially from those previous studies. But the sections developed by the authors should be highlighted. In this way, the readability will be improved and the novelty will be highlighted.*

We agree, that a rearrangement and tightening of the model description section would be helpful especially as the comments of the other reviewer go into the same direction. We have fixed this in the revised version of the manuscript by shortening the description of the original linear model and rearrange Section 2: General Description, Description of the original approach, modifications made by us and a Section on pre-preparations and general simulation procedure (flowchart).

*Another aspect is not so clear. The procedures for extreme events do not include a component for time series generation. For weather generator, it usually generates time series of wet events. However, it appears that the current procedure develop the PDFs of different seasons based on a group of extreme wet events, and then used to generate another group of extreme events. If so, how to apply the results for future risk assessment?*

You are right, based a seasonal PDFs of historic extreme events, a stochastic event set is computed with a certain number of at first independent events. It is possible to estimate a corresponding somehow continuous time period as it is described in Section 6: Counting the number of events in the stochastic event set exceeding a defined threshold, for example, in our case the 99$^{th}$ percentile of the observations (regarding spatial mean precipitation) and normalize it with its probability. The resulting total time span $T_{SPM}$ can be used to estimate the return period of every single event (like in Figure 15), to rank the events, and to estimate a new PDF (Gumbel) which can be used for the risk assessment. The rainfall events can be used as input for hydrological rainfall-runoff-models. We have added a statement in the conclusions on how to transfer the results to future risk assessments.

**Point-by-point response to Reviewer #4**

Florian Ehmele on behalf of the co-author

December 19, 2018

Thank you very much for your work and the useful and valuable comments that helped to improve the scientific quality of our manuscript. Please find below our reply to the individual points.

Dear Florian Ehmele and Michael Kunz,
I have now finished reading your manuscript entitled "Flood-Related Extreme Precipitation in Southwestern Germany: Development of a Two-Dimensional Stochastic Precipitation Model". I found the manuscript interesting and I think the hydrological community will benefit from the advancements in Smith and Barstad's stochastic rainfall model you are suggesting. The manuscript fall within the topics cover in HESS, and I believe that after revising the text the paper can be accepted for publication.

Thank your very much again for you positive feedback on the general topic of the paper.

Saying that, I have struggled to read the paper till the end. In its present form the text is hard to read and to follow. My main comments are therefore related to the structure and length of the text:

- The text is too long. I suggest moving some of the Tables and Figures to the supplementary material (SI) – see some suggestions in the specific comments below. The same goes to the text itself – some of the descriptions can be summarized in a Table (list of model parameters, for example) and some can be moved to the SI. In addition, the text can be shorten in many places. For example, Section 3 (data sets) could be summarized in a single page (2.5 pages currently). The first sub-section describing the rainfall product can be summarized in two sentences, refereeing to REGNIE and indicating the product limitations.

After rereading the paper again, we agree that at many places the text can be tightened. Saying that we tried to rephrase the text without loosing to much necessary information. We follow the suggestion to move some of the Figures to a supplementary document, and we moved some information to an appendix.

- There are many replicas in the text that need to be eliminated. For example, Section 2.6 "Stochastic modeling of precipitation events with SPM2D requires the adjustment of appropriate probability density functions (pdfs) to all input parameters..." and in the beginning of Section 5.1 "Stochastic model simulations are based on pdfs that are adjusted to the required parameter...". The text should be concise as possible, especially when the text is of technical nature like in this manuscript.

We carefully checked the paper again for replicas and eliminated them as far as possible.

- Some restructuring is needed. In many cases I thought that information is missing and that the text is incomplete just to realize that the information I looked for is written later in the text. For example, in page 5 line 15 you mention that "R may become negative" and only later (page 6 line 25) you explain that even if it is negative you consider it to be zero. Another example, the model is briefly described at the introduction and then again in Section 2.1. I suggest to follow a simple structure: general introduction, description of the model component, model calibration (general calibration procedure, not tailored to the case study, so readers can understand what input is required and how to calibrate the model), short description of the study area, short description of the data sets, calibration and results of the case study, and conclusions.

We also agree in this point with the reviewer and tried to rearrange the text that information is given at the right place (see also specific comments below). We removed the mode description from the introduction to keep it general. As a restructuring of Section 2 was also recommended by the other reviewer we applied it as follows: General Description of the complete stochastic model (2.1), brief description of the underlying linear orographic model by Smith and Barstad with its components (2.2), description of the modifications implemented in this study (2.3), and pre-preparations and general simulation procedure (2.4). We now think that our modifications are more highlighted and the overall operating principles became more clear. At this point we need to clarify that the given calibration in Sect. 4 is the general procedure using the training sample of historic extreme events and the case study (4.4) is an example (out of that sample) of the model output after calibration.

Below please find my specific comments to the text. I hope that you will find my comments useful and take my criticism positively. I value your study and I hope they will assist in improving the manuscript.

Again many thank, your comments indeed have been very helpful.

Specific comments
[page line]

[2 1] Actually, IDF curves can be estimated over large areas using gridded information from remote sensing. In recent years there were several advances in this direction. Search, for example, for recent publications by Francesco Marra.
[2 4-6] For more recent studies were rainfall generators were used to compute spatially distributed IDF curves I can suggest the Authors to also consider two recent papers that we (Peleg et al., 2017 and 2018) have published that address this topic explicitly.
Peleg et al. (2017). Partitioning the impacts of spatial and climatological rainfall variability in urban drainage modeling. Hydrology and Earth System Sciences, 21(3), pp.1559-1572.
Peleg, N. et al. (2018). Spatial variability of extreme rainfall at radar subpixel scale. Journal of Hydrology, 556, pp.922-933.

Thanks for mentioning this new and interesting studies. We updated the introduction concerning this studies.

[2 14-25] This paragraph belongs to the method section, where the model is described.

Moved to and included in Section 2.1.

[4 3] Why only stratiform clouds? I see in eq. (1) that the convection component is also consider. I suspect that Eq 1 is not referring to the original model but to the developments you added later. Maybe first describe simply the original model as it, and then added a section explaining the changes you are suggesting.

The SPM2D is designed for widespread heavy precipitation related to pluvial flood events for which convection is of minor importance. Nevertheless convection can appear atop stratiform clouds influencing the very local precipitation distribution. This was the reason for us to implement this term. Equation 1 gives the total precipitation as it is calculated in the complete SPM2D and includes the original Smith-Barstad model as well as our modifications. As mentioned to the general comments, we rearranged Section 2 completely, so this should become clear.

[4 7] "large scale lifting". What do you mean by that? upward lifting (i.e. omega component)? Isn't that part of vertical convection?

We rephrase the according text for clarification. Large-scale lifting is referred to the omega-equation with its components of vorticity advection, warm air advection and diabatic phase transition which lead to large-scale vertical motions. Conversely, convection (in meteorology) describes small-scale thermally driven circulations (e.g. buoyancy) which is a different type of physical process, and so we differentiate it accordingly.

[4 9] "rSPM". If I understand right, you are modifying the original SPM2D model to be a reduced complexity model for some components while adding additional components that were not part of the original model. The fact that you have two names for the same model (SPM2D and rSPM) is confusing. Mention the changes made to the model (can also be given in a form of a table – original model components and the new model components) so it will be clear what is what.

SPM2D is the new complete model, rSPM is meant as the original Smith-Barstad model. Along with the restructuring we changed the name to SBM to separate it more distinctly. We also reduced the number of acronyms (see further comments below).

[4 9] "we included two additional precipitation components". Which are not part of the original Smith and Barstad model? If this is the case, then equation 1 should be with only the components originally used in SPM2D and a new equation 2 is to be given with the components of rSPM. That will also resolve my above comment related to the stratiform cloud.

This should be clear now with the new text structure.

[4 11-13] The sentence is not that clear, please rephrase.

We fixed this.
[general comment] Please add a table that summarize the parameters of the model.

After tightening and rearranging the text, we think an additional table is not necessary, especially as Table 2 (Sect. 5.1) gives this overview when summarizing the results of the pdf estimation which is the required part to run the SPM2D in stochastic mode.

[4 15] "is simulated". Simulated how? For each year the total numbers of events and the length for each events are sampled from a given distribution? If so, which distribution is used? Please give more details here.

This should be clear now in the Sect. 2.4.2 giving a description of the general simulation procedure. A total number of independent events (here 10,000) is simulated using the estimated seasonal pdfs. Which type of pdf best fits the observed distribution for each variable is given later in the results part (i.e. Table 2, Sect. 5.1) and not of importance at this point. The stochastic event set than can reallocated to a certain time period as described in Section 6, which then is useful e.g. for risk assessments.

[4 18-19] I am missing some information here. What is the spatial and temporal resolution of the model? Is the model daily or sub-daily?

This should now be clarified within the introductory paragraph to Section 2 and Section 2.4.2. The spatial resolution of the model is 1 km² in accordance to the mainly used REGNIE observations. The temporal resolution of the SPM2D is 24 hours. Within every time step, the stochastic precipitation parts of fronts and convection are calculated once and treated as footprints in daily precipitation. The SBM parts of orographic and synoptic background precipitation are calculated twice in

accordance to the 12-houly soundings we used and to account for slightly variations of the ambient conditions during the day. We add this comment to the text.

[4 30] "sounding data". Can also be estimated from other sources, such as reanalysis data? A matter for a latter discussion.

We use sounding data in this study, but yes, vertical profiles can be estimated from reanalysis data as well. We add a comment on that to the conclusions.

[Figure 2] What are all the larger/smaller values from zero refer to? For example, for Roro, which arrow is for values larger than zero and which arrow to follow when values are smaller than zero? Rtot will always be larger than zero as only wet events are consider, no?

We add two additional arrows for the cases of Roro < 0 and Rfront < 0 to Fig. 2. With the restructured text and the answers to previous comments, this should also be clear now. Both Roro and Rfront can attain negative values in descent regions due to evaporation. These negative values are taken into account in the summation of Rtot. And the end, values of Rtot <0 are truncated away as it is non-physical. Therefore, Rtot can reach zero at some grid points but regrading spatial means only wet events are simulated with SPM2D.

[5 9] I can understand why the time sacles are constant in time, but why they are constant in space?

This is an assumption of the original Smith-Barstad model and one reason why the extent of the model domain is limited. The time scales are assumed to be homogeneous regrading mesoscale areas. We add a comment on that in the text.

[5 15] And is consider negative when embedded in equation 1 or is set to zero?
[6 24-26] OK, that answer my previous question. This should have been mentioned earlier.

See above reply on the comment on Figure 2. Fixed in the text.

[7 4] I do not see it in equation 6. Did you meant eq. 7?
[7 11] Eq. 7. Equation 6 is still on the k,l plane.

We combined the former Equations (4) and (6) to a final function for orographic precipitation. The new Equation (4), then, visibly contains all the modifications made. With that, the text should be better understandable.

[7 15] "whereas Coro is constant over the whole domain". From Fig. 3 it seems that Coro is changing along the x-y plane and is not constant over the whole domain. Is it constant in time? I guesses you meant here to say that Coro is affecting all grid cells in all times.

Coro is a constant number which affects all grid points and throughout all events which reduced the orographic part by a certain degree. In regions with high values of Roro the effect of Coro is more pronounced than in regions where Roro is of less importance. We modified the according text for better understanding.

[8 2-4] This explanation should be moved up, after equation 1 is introduced.

Fixed.

[8 30] The value of Cfront is determined from a distribution? Or is it constant for each season? Depend on the wind direction? Please add information on how the values of Cfront are defined.

As described in the text, a front is implemented as a rectangular area with infinitely extent along the front-parallel axis and a Guassian-shaped smoothing along the front-normal axis (we also adjusted the notations in Figure 4). The each time step, the maximum of this distribution is given by Cfront using the corresponding seasonal differentiated pdfs and than smoothed to the borders of the rectangular. We modified the text for better understanding. Furthermore, Cfront is independent from wind direction.

[9 10] "Note, however, that the model is not foreseen to simulate purely convection". But, often the most extreme rainfall is (almost) purely convective in nature - isn't that contradict the purpose of this study, to explicitly account for the extreme (200-y return period) rainfall events?

You have to separate between pluvial flood events and flash floods. Flash floods are driven by almost purely (free) convective precipitation. Pluvial floods along the major rivers, which are the focus of this study, are not related to free convection but are caused by widespread long-lasting precipitation from stratiform clouds where convective cells can be include leading to locally enhanced rainfall totals. This is the reason why we add a convective precipitation part Rconv, but it is different from (free) deep moist convection. We modified the text for better understanding.

[10 9] "convective cells". It makes sense to follow the cell structure only if the temporal resolution of the model is hourly or sub-hourly (is it? reading so far I couldn't find the information about the temporal resolution of the model), with coarser resolution you cannot detect cells' structure anymore.

As mentioned above the SPM2D is not foreseen to simulate purely convection in detail. The expression "convective cell" might be inappropriate in this case, so we changed it to "footprint" which is more of what was the intention – regions of enhanced precipitation due to convection visible in daily totals.

[10 12] I get from this that the grid cell size is 1 km? That information should be written somewhere.

We add a short paragraph on that at the beginning of Section 2.

[11 3] "the spatial distribution of Cconv randomly varies between the given limits". Why? Anyhow you do not try to capture the spatial structure of the cells, so why adding another level of complexity?

Even though convection is simplified and seen as footprints in daily precipitation, it is adequate to vary Cconv in between such a footprint instead of assigning a constant value for the entire footprint area. In an extreme case a footprint can be of 300km length and in order of several 10km width. Assigning a constant value to this large area would produce a less realistic precipitation distribution, so we vary Cconv and smooth it at the end. Nevertheless, for potential subsequent hydrological simulations widespread precipitation is of greater importance and the difference between a fixed or a varying Cconv is negligible.  We add a comment to the text.

[12 2] March-April-May, MAM from hereon. Same for the other seasons.

Fixed.

[Table 1] Instead of list of distributions that were examined, I suggest having a table summarizing the distribution used for each of the variables.
- Further reading, I see that this information is given in Table 3. I thus do not see the need in having Table 1 (maybe as supplementary material) and suggest to remove it.

We agree and put Table 1 into an appendix section as it is useful for the pdf acronyms.

[16 6] r refer here to the spatial correlation of a given rain field?

Despite putting the description of the skill score to the appendix section, we changed the text to clarify this comment. r in this case is the Spearman correlation coefficient between the observed and simulated precipitation field (2D).

[Figure 7] Can be moved to the supplementary material.
[Figure 9] can be moved to the SI as well.

As recommended we put both figures into a supplementary document.

[21 7-11] There is more than the mean areal rainfall that needs to be consider when comparing maps of extreme rainfall. For example, how well the structure (dimension and location) of the heavy rainfall (e.g. above 50 mm per day) are reproduced?

Such information, actually, is given at the beginning of the case study when describing the spatial distribution of the observed and simulated precipitation filed. To pronounce it a little bit more we add hard numbers of overall maximum precipitation and the area exceeding R=50mm.

[26 7] Simulated how? I guess that for a given event the parameters were sampled from the relevant distributions independently. Are the model variables (e.g. the lapse rates) corss-correlated and, if so, is the cross-correlation accounted for in the model (for example, by sampling the parameters from the pdfs using copulas). Please provide more information here.

The simulation procedure is given in Sect. 2.4.2. but we also add some information at this point: Every single event of the 10,000 has a specific duration leading to a total number of approx 31,500 days. For each of the input variables a vector of values of the same length (31,500) is estimated using the corresponding pdfs giving the required input for each day. The variables are treated as independent. A correlation analysis was given in Sect. 5.1 and reveals less correlations so that a coupling of pdfs via copulas is not necessary.

[new table] I suggest adding a new table summarizing the different models and experiments (rSPM10k, CCLM, rSPM…). It becomes difficult to follow all the names.

During the revisions we conclude that some of the acronyms are not necessary or not any more. Therefore, we reduced the number of names to a minimum so that an extra table is not required.

[Figure 16] SI

Done.

[revised manuscript text omitted]

---

## Author Response (AR3)

**List of Major Changes**

(1) Adjustments to the introduction
- Brief statement on other currently available 2-D stochastic weather generators
- Highlighting the novelty of the presented approach

(2) Adjustments to the conlcusions
- Comment on the novelty of the presented approach compared to other 2-D stochastic weather generators

**Point-by-point response to Reviewer #4**

Florian Ehmele on behalf of the co-author

February 06, 2019

Thank you very much again for your work and the useful and valuable comments that helped to improve the quality of our manuscript. Please find below our reply to the individual points.

Dear Authors,

The current version of the manuscript is much improved in comparison to the original version. The Authors did reply to all my comments satisfactory.

We are happy to hear, that we could answer all your statements.

From my side, there is only one minor issue that remain unsolved. The paper present a new stochastic gridded rainfall model, and the motivation for it is well described - the need to generate multiple realizations to address extreme rainfall events. However, the Authors ignore the fact that there are many weather generators, stochastic and 2-dimensional, which are already available. This is explicitly written in Page 2 Line 10-13: "... Albeit considering the long-term variability of precipitation, which leads to more reliable estimates for extremes, these approaches still lack spatial representativeness". This statement is simple not true. There are several models that produce the same output as your model, and even at much higher space-time resolution (sub-daily, sub-kilometer), to name a few: STORM (Singer et al., 2018), STREAP (Paschalis et al., 2013) that is an advanced version of the "String of Beads" models (Pegram and Clothier, 2001), AWE-GEN-2d (Peleg et al., 2017) and a model recently presented by Benoit et al. (2018).

This is indeed a very useful and important comment. To be honest we didn't do our literature work carefully in that point looking just for physically based weather generators and did not include some statements on generally available stochastic weather generators. Also we didn't check for the latest work on this topic. Thanks again of mentioning this, we will definitely include something related to this.

I suggest the Authors to acknowledge the fact that there are other stochastic 2-dimensional rainfall models in the market, and to explicitly discuss the advantages of the model suggested here in comparison to the others. As I see it, the model suggested by the Author is a semi-physical model, thus can be consider more "robust" than the other stochastic models I listed abode that are statistically based. I suggest the Authors to mention something in this direction at the introduction and conclusions sections.

To deal with this important comment, we extend the paragraph on stochastic models in the introduction section with few statements on the studies mentioned above. Furthermore, we relate our study in the scope of this topic and highlight the novel physically based approach. We adjust the conclusions section with a comment on that.

Other than that I think the manuscript is ready for publication.

Thank you very much.

[revised manuscript text omitted]